# Training High Performance Spiking Neural Network by Temporal Model Calibration

**Jiaqi Yan** [1 2]  **Changping Wang** [1 2]  **De Ma** [1 2]  **Huajin Tang** [1 2]  **Qian Zheng** [1 2]  **Gang Pan** [1 2]

## Abstract

Spiking Neural Networks (SNNs) are considered promising energy-efficient models due to their dynamic capability to process spatial-temporal spike information. Existing work has demonstrated that SNNs exhibit temporal heterogeneity, which leads to diverse outputs of SNNs at different time steps and has the potential to enhance their performance. Although SNNs obtained by direct training methods achieve state-of-the-art performance, current methods introduce limited temporal heterogeneity through the dynamics of spiking neurons or network structures. They lack the improvement of temporal heterogeneity through the lens of the gradient. In this paper, we first conclude that the diversity of the temporal logit gradients in current methods is limited. This leads to insufficient temporal heterogeneity and results in temporally miscalibrated SNNs with degraded performance. Based on the above analysis, we propose a Temporal Model Calibration (TMC) method, which can be seen as a logit gradient rescaling mechanism across time steps. Experimental results show that our method can improve the temporal logit gradient diversity and generate temporally calibrated SNNs with enhanced performance. In particular, our method achieves state-of-the-art accuracy on ImageNet, DVSCIFAR10, and N-Caltech101. Codes are available at https://github.com/zju-bmi-lab/TMC.

## 1. Introduction

Spiking Neural Networks (SNNs), unlike traditional Artificial Neural Networks (ANNs), mimic the dynamic behaviors of biological neurons and have spatio-temporal information process capability via spike transmission (Maass, 1997; Hu et al., 2021; Li et al., 2024a). The rise of neuromorphic computing (Ma et al., 2024; Davies et al., 2018) has allowed SNNs to achieve enhanced performance and greater energy efficiency, garnering extensive attention (Yan et al., 2024; Liu et al., 2024; Hu et al., 2023) and driving widespread applications (Liao et al., 2024; Zhang et al., 2024; Yan et al., 2025). Previous works have demonstrated that neuronal dynamics lead to temporal heterogeneity in SNNs (Perez-Nieves et al., 2021; Chakraborty & Mukhopadhyay, 2024; Wang et al., 2025; Chen et al., 2025). This characteristic enables SNNs to extract temporal dynamic features and produce temporally extended diverse outputs that are transformations of those features (Wu et al., 2025; Gast et al., 2024; She et al., 2021b; Savin & Denève, 2014). By integrating the diverse outputs to make decisions, the performance of SNNs can be further improved, especially in tasks with complex temporal structures, such as neuromorphic datasets (Gast et al., 2024; Zheng et al., 2024; She et al., 2021b; Chakraborty & Mukhopadhyay, 2023).

The rate-coding based SNNs trained using direct training methods based on BackPropagation Through Time (BPTT) and surrogate gradient (Wu et al., 2018; Deng et al.) have achieved state-of-the-art performance (Zhou et al., 2024; Yao et al.). To further explore the potential of SNNs, researchers focus on enhancing the temporal heterogeneity of the rate-coded SNNs during direct training from two perspectives. The first category (Rathi & Roy, 2021; Fang et al., 2021b; Yao et al., 2022) increases the heterogeneity of neurons by proposing parametric spiking neurons. They set one or more membrane-related parameters learnable to enlarge the representation space and improve the performance of SNNs. The second category (She et al., 2021b;a) investigates the relationship between heterogeneity and network structure and demonstrates that recurrent connections are beneficial to capture different features of temporal patterns.

However, gradient descent in the temporal dimension determines the learning of temporal features (Wu et al., 2018; Huh & Sejnowski, 2018), and how to enhance the temporal heterogeneity of SNNs through the lens of gradient has not been fully studied. Following existing works (Huh & Se-

[1]The State Key Lab of Brain-Machine Intelligence, Zhejiang University [2]College of Computer Science and Technology, Zhejiang University. Correspondence to: Qian Zheng <qianzheng@zju.edu.cn>, Gang Pan <gpan@zju.edu.cn>.

jnowski, 2018; Perez-Nieves & Goodman, 2021), the gradients in direct training methods of SNNs can be decomposed into two components: the temporal gradients propagated backward through hidden layers (temporal hidden layer gradients) and the temporal gradients of the loss function with respect to the logits (temporal logit gradients). The temporal hidden layer gradients, determined by the spiking neuron behaviors, have been shown to exhibit temporal heterogeneity (Perez-Nieves & Goodman, 2021). Nevertheless, the temporal logit gradients, which are derived from the loss function and calculated as the error between model outputs and the optimization target across time steps, have not been sufficiently investigated.

In this work, we investigate the temporal logit gradients under existing loss functions and conclude that they have insufficient diversity in the temporal dimension. We demonstrate that this is not beneficial to capture temporal dynamic features across time steps, limits the temporal heterogeneity, and results in temporally miscalibrated SNNs with degraded performance. To address this, we introduce the concept of model calibration in ANNs (Mukhoti et al., 2020; Ross & Dollár, 2017; Tang et al., 2020), and propose a Temporal Model Calibration (TMC) method. TMC can be seen as a temporal gradient rescaling mechanism to generate diverse logit gradients in the temporal dimension. Our main contributions can be summarized as follows:

- We analyze the temporal logit gradients calculated in existing direct training methods and conclude that they have insufficient diversity in the temporal dimension. This restricts temporal heterogeneity and leads to miscalibrated SNNs with degraded performance.

- Inspired by the concept of model calibration in ANNs, we propose the TMC method, which can be seen as a temporal logit gradient rescaling mechanism.

- Extensive experimental results demonstrate that the proposed method can improve the temporal logit gradient diversity and generate temporally calibrated SNNs that exhibit enhanced performance. It is worth noting that our method achieves state-of-the-art performance on ImageNet and neuromorphic datasets.

## 2. Related Work

**Temporal Heterogeneity of SNNs.** As the basic information processing units, spiking neurons exhibit rich dynamic behaviors and determine the intrinsic properties of SNNs, which are different from ANNs (Wu et al., 2018; Roy et al., 2019). Existing work has shown that neuronal dynamics lead to temporal heterogeneity in SNNs (Perez-Nieves et al., 2021; Chakraborty & Mukhopadhyay, 2024; Wang et al., 2025). Experimental evidence shows that temporal heterogeneity enhances the performance of SNNs, particularly

in tasks with complex temporal structures, such as neuromorphic datasets (Gast et al., 2024; Zheng et al., 2024; She et al., 2021b; Chakraborty & Mukhopadhyay, 2023). Furthermore, the relationship between temporal heterogeneity and the performance of SNNs has been widely studied. Recent work demonstrates that temporal heterogeneity enables SNNs to extract dynamic features and produce temporally diverse outputs that are transformations of those features (Wu et al., 2025; Gast et al., 2024; She et al., 2021b; Savin & Denève, 2014). Based on rate coding, making decisions by integrating the diverse outputs improves the performance of SNNs.

**Direct Training Strategies for SNNs.** To enhance the performance of SNNs, existing direct training methods introduce training strategies from both forward and backward perspectives. On the one hand, to enlarge the representation space of SNNs, current studies (Rathi & Roy, 2021; Fang et al., 2021b; Yao et al., 2022) propose different parametric spiking neurons, which set one or more membrane-related parameters to be learnable during training. In addition, researchers (She et al., 2021b;a) introduce recurrent connections to approximate temporal mapping functions. On the other hand, current works propose different loss functions to determine the temporal optimization of SNNs. Specifically, Standard Direct Training method (Wu et al., 2018) uses the average membrane potential of the last layer as the final output and optimizes it towards the one-hot label. The Temporal Efficient Training (Deng et al.) method assigns the true label as the optimization target for each time step. Recently, new loss functions have been proposed (Dong et al., 2024; Zhao et al., 2025; Qiu et al., 2024; Zuo et al., 2024), which introduce additional optimization constraints of outputs based on SDT.

**Model Calibration in Deep Learning.** Recent evidence has shown that deep networks are prone to make overconfident predictions due to miscalibrated output probabilities (Mukhoti et al., 2020; Müller et al., 2019). To mitigate the issue, two main families of approaches have emerged recently: post-processing methods (Tomani et al., 2021; Ma & Blaschko, 2021) and learning-based methods (Ovadia et al., 2019; Szegedy et al., 2016; Mukhoti et al., 2020). Nevertheless, post-processing methods do not work well under data distribution shifts (Ovadia et al., 2019). Thus, learning-based calibration methods have become a more powerful choice. Muller et al. (Szegedy et al., 2016) propose Label Smoothing on soft targets, which aids in improving calibration. Mukhoti et al. (Mukhoti et al., 2020) showed that focal loss (Ross & Dollár, 2017) can implicitly calibrate models by reducing the KL-divergence between predicted and target distribution. Recently, common learning-based calibration methods use additional loss terms as a regularization term (Chen et al., 2024; Pereyra et al., 2017), which has achieved state-of-the-art performance. It is notable that

learning-based model calibration methods fundamentally have the rescaling effect to the logit gradient (Tang et al., 2020; Lee et al., 2022).

## 3. Method

### 3.1. Problem Definition

**Temporal Heterogeneity Analysis.** Here, we provide the analysis of the temporal heterogeneity within a spiking neuron under both continuous and discrete time.

*Continuous Time.* The behavior of a spiking neuron over continuous time can be formulated as follows:

$$\tau \frac{du_t}{dt} = -(u_t - u_{reset}) + I_t, \quad u_t < V_{th}, \quad (1)$$

$$o_t = \sum_{t_f} \delta(t - t_f), \quad (2)$$

where $u_t$ is the membrane potential, $\tau$ is the time content, $I_t$ is the input current, and $V_{th}$ is the threshold. $\delta(\cdot)$ is the Dirac delta function. The spike train $o_t$ is generated when $u_t$ reaches $V_{th}$ at time $t_f$, and $u_t$ is reset to the resting potential $u_{reset}$.

In Equation A.5, the change of $u_t$ is quantified by $\frac{du_t}{dt}$, which is determined by two components $-(u_t - u_{reset})$ and $I_t$. The term $-(u_t - u_{reset})$ reflects the natural decay of the membrane potential back toward the resting potential $u_{reset}$. Meanwhile, $I_t$ captures the influence of the temporally varying external current on the membrane potential and spike firing. Overall, $|\frac{du_t}{dt}|$ is likely to be greater than zero, indicating ongoing changes in the membrane potential and spike train. Notably, for complex temporal structures, such as those found in neuromorphic datasets, a higher degree of temporal heterogeneity is necessary to effectively capture dynamic features. More detailed analysis can be found in Appendix B.

*Discrete Time.* To establish the computational link along the spatial-temporal dimension, a discrete-time spiking neuron model is utilized for constructing SNNs as follows:

$$u_{t+1}^i = (1 - \frac{1}{\tau})(u_t^i - V_{th}o_t^i) + \sum_j W^{ij}o_{t+1}^j, \quad (3)$$

$$o_t^i = H(u_t^i - V_{th}), \quad (4)$$

where $\mathbf{W}$ is the trainable synaptic weights, and $H(\cdot)$ is the Heaviside step function.

In Equation 3, the change of $u_{t+1}^i$ is determined by $(1 - \frac{1}{\tau})(u_t^i - V_{th}o_t^i)$ and $\sum_j W^{ij}o_{t+1}^j$. The first term, that is, the neuron's intrinsic temporal membrane potential, can be enriched by incorporating one or more trainable parameters related to the membrane, which has been explored in current studies (Rathi & Roy, 2021; Fang et al., 2021b; Yao

et al., 2022). The second term, which represents the temporal input current, can be enriched through two primary approaches: Diversifying the input spike train $o_t^i$ through recurrent connections, which has been investigated in current studies (She et al., 2021b;a). Expanding the representation space of $\mathbf{W}$. However, this approach is a promising yet under-researched approach. In the following section, we present a comprehensive analysis of the temporal training dynamics for $\mathbf{W}$.

**Temporal Gradient Analysis.** In the direct training methods, SNNs are updated by computing the gradient of the loss function with respect to the synaptic weights in each layer. Using Backpropagation Through Time (BPTT) with surrogate gradients (Wu et al., 2018), the weight gradients can be calculated as:

$$\frac{\partial L_{CE}}{\partial \mathbf{W}} = \frac{1}{T} \sum_{t=1}^{T} (\boldsymbol{\alpha}_t - \boldsymbol{Y}_t) \frac{\partial \mathbf{Z}_t}{\partial \mathbf{W}}, \quad (5)$$

where $L_{CE}$ represents the cross-entropy (CE) loss. At time step $t$, $\boldsymbol{\alpha}_t$ is the optimization objective, which is determined by the form of loss functions, and $\boldsymbol{Y}_t$ is the optimization target. The synaptic weights $\mathbf{W}$ are used to process the given input and produce the logit $\boldsymbol{Z}_t$. $T$ is the total number of time steps.

In Equation (5), the update gradient of $\mathbf{W}$ at each time step can be decomposed into two terms. The first term $\frac{\partial \mathbf{Z}_t}{\partial \mathbf{W}}$ is the gradients propagated backward down to the input layer across time steps (temporal hidden layer gradient). It is associated with the dynamic behaviors in spiking neurons and have been proven to be heterogeneous in the temporal dimension (Perez-Nieves & Goodman, 2021). The second term $(\boldsymbol{\alpha}_t - \boldsymbol{Y}_t)$ is the partial derivative of the loss function to the logit over time steps (*temporal logit gradient*). It is calculated as the error between the model's outputs and the true labels, determining the optimization objective of SNNs. To effectively represent temporal dynamic features, $(\boldsymbol{\alpha}_t - \boldsymbol{Y}_t)$ should provide a reasonable optimization magnitude and direction of $\mathbf{W}$ at each time step while maintaining temporal diversity across time steps. However, as demonstrated in the following section, the temporal logit gradients in current direct training methods don't exhibit these properties simultaneously.

**Standard Direct Training (SDT)**, as defined by Deng et al. (Deng et al.), calculates the cross-entropy loss between the average pre-synaptic input of the output layer and the true label. For SDT, the temporal logit gradient [1] $\nabla \boldsymbol{Z}_t$ at time step $t$ can be derived as:

$$\nabla \boldsymbol{Z}_t^{\text{SDT}} = (\boldsymbol{P}_{\text{avg}} - \boldsymbol{Y}), \quad (6)$$

---

[1] Formally, the temporal logit gradient is defined as the partial derivative of the loss function with respect to the logit output $\boldsymbol{Z}_t$, or $\frac{\partial L}{\partial \boldsymbol{Z}_t}$ in the mathematical form. In the rest of the paper, we will use the symbol $\nabla \boldsymbol{Z}_t$ for its simplicity.

where $\boldsymbol{P}_{\text{avg}}$ is the softmax probability of the average output logit and $\boldsymbol{Y}$ is the one-hot encoded training label. Given that $\nabla \boldsymbol{Z}_t^{\text{SDT}}$ is the same across time steps, it constrains the gradient diversity within SDT. Recently, new loss functions have been proposed in (Dong et al., 2024; Zhao et al., 2025; Qiu et al., 2024; Zuo et al., 2024). They introduce additional loss terms as a regularization term, which can be regarded as variants of SDT. Their objective is to realize the consistent representation of SNNs, which is not compatible with the temporal heterogeneity of SNNs.

**Temporal Efficient Training (TET)** (Deng et al.) first calculates the cross-entropy loss between the pre-synaptic inputs of the output layer with the true labels at each time step. Then, it employs the mean of these losses as the final loss. For TET, the temporal logit gradient is expressed as:

$$\nabla \boldsymbol{Z}_t^{\text{TET}} = (\boldsymbol{P}_t - \boldsymbol{Y}), \tag{7}$$

where $\boldsymbol{P}_t$ is the softmax probability of the output logit at time step $t$. Although the logit gradients have different magnitudes across time steps due to varying $\boldsymbol{P}_t$, they homogenously guide the optimization of temporal outputs in the same direction toward the one-hot encoded training label $\boldsymbol{Y}$. However, existing studies have presented that minimizing the CE loss optimizes $\boldsymbol{P}_t$ to infinitely match the training label $\boldsymbol{Y}$. This results in temporally miscalibrated SNNs with overconfidence issues and limited performance (Mukhoti et al., 2020; Müller et al., 2019).

### 3.2. Gradient Rescaling in Model Calibration

Based on the above analysis, the temporal logit gradients should provide a reasonable optimization magnitude and direction of $\mathbf{W}$ at each time step while maintaining temporal diversity across time steps to generate temporally calibrated SNNs. In the following sections, we explicitly define the temporally calibrated SNNs and propose the temporal logit gradient rescaling mechanism that can generate the SNNs.

**Temporally Perfectly Calibrated SNN.** At each time step, consistently optimizing the predictive output $\boldsymbol{P}_t$ towards the one-hot encoded training label $\boldsymbol{Y}_t$ achieves impressive performance and is commonly used in the direct training of SNNs. However, based on the concept of model calibration, this approach leads to an overfitting issue and generates miscalibrated SNNs with limited performance. To address this issue, we first introduce the definition of a perfectly calibrated model in ANNs and then extend the definition to the temporal dimension for SNNs.

Specifiaclly, a model is perfectly calibrated if and only if the predicted probability confidence of each sample equals the model's accuracy. This can be mathematically expressed as $\hat{P} = \mathbb{P}(\hat{y} = y|\hat{P})$, where $\hat{P}$ is the confidence, that is, the highest value within the predictive probability $\boldsymbol{P}$. $\hat{y}$ denotes the predicted class, and $y$ denotes the true label. We further extend the definition to the temporal dimension for SNNs, which is defined as follows:

**Definition 3.1.** At time step $t$, an SNN is temporally perfectly calibrated if and only if:

$$\hat{P}_t = \mathbb{P}(\hat{y} = y|\hat{P}_t), t \in \{1, 2, \ldots, T\}. \tag{8}$$

As our work focuses on the rate-coded SNNs that currently achieve state-of-the-art performance, we incorporate specialized analysis for these SNNs based on Definition 3.1. Specifically, a rate-coded SNN makes decisions according to the average accumulated temporal information. We use $\bar{\boldsymbol{Z}}_t = \frac{1}{t} \sum_{i=1}^{t} \boldsymbol{Z}_i$ to refer to the average accumulated logit at time step $t$, where $t \in \{1, 2, \ldots, T\}$. Then, the temporal confidence $\hat{P}_t$ can be represented by the maximum value of Softmax($\bar{\boldsymbol{Z}}_t$). Notably, Li et al. (Li et al., 2024b) suggest that the accuracy of rate-coding SNNs should increase monotonically with time steps if temporal heterogeneity exists. Based on this observation, the confidence of a temporally perfectly calibrated SNN should increase monotonically with time steps, and we have the following remark:

*Remark* 3.2. Given a rate-coded SNN with a $T$ time steps, let $\hat{P}_t$ denote the predicted probability confidence at time step $t$. If the SNN is temporally perfectly calibrated, then for any $t$, $t = 1, ..., T-1$, it satisfies $\hat{P}_t < \hat{P}_{t+1}$.

**Temporal Logit Gradient Rescaling.** To generate the temporally perfectly calibrated SNNs, we draw inspiration from the learning-based model calibration methods. These methods introduce novel loss functions built upon the CE loss, demonstrating the most effective performance for model calibration in ANNs (Chen et al., 2024; Pereyra et al., 2017). We analyze the optimization effect of the loss functions in the learning-based model calibration methods, and then extend the effect to the temporal dimension for direct training of SNNs.

Specifically, the loss functions in current learning-based model calibration methods realize calibration during optimization by applying the rescaling effect to the logit gradient of the CE loss. This can be formulated as:

$$\nabla \boldsymbol{Z}^* \oslash \nabla \boldsymbol{Z}^{\text{CE}} = g(\boldsymbol{P}), \tag{9}$$

where $\oslash$ is the element-wise division operator, $\boldsymbol{Z}$ is the output logit, $\nabla \boldsymbol{Z}^*$ is the logit gradient of the loss function in the model calibration method, and $\nabla \boldsymbol{Z}^{\text{CE}}$ is that of CE loss. $\boldsymbol{P}$ is the softmax probability of $\boldsymbol{Z}$, and the highest value within $\boldsymbol{P}$ denotes the predicted probability confidence. $g(\cdot)$ represents the rescaling function, which is influenced by the confidence and is adaptively diverse between different samples and training phases. The confidence-related rescaling effect on the logit gradient of $\nabla \boldsymbol{Z}^{\text{CE}}$ finally produces the model-calibrated logit gradient $\nabla \boldsymbol{Z}^*$. Thus, the

model calibration methods in ANNs can adaptively generate model-calibrated logit gradients and alleviate the overfitting issue where the predicted probability confidence is infinitely close to 1, ignoring the model accuracy.

With further instantiation specific to the rate-coding SNNs, we extend the logit gradient rescaling effect to the temporal dimension for direct training of SNNs. We denote the temporal rescaling factor as $g_t$ and define it as:

**Definition 3.3.** Let $\nabla \mathbf{Z}_t^*$ be the temporal logit gradient of the loss function in the temporal model calibration method. Similar to Equation 9, $g_t$ is defined as:

$$\nabla \mathbf{Z}_t^* \oslash \nabla \mathbf{Z}_t^{\text{CE}} = g(\mathbf{P}_t | \mathbf{Z}_1, \mathbf{Z}_2, ..., \mathbf{Z}_t). \tag{10}$$

Notably, due to the rate coding mechanism of SNNs, $g_t$ is required to satisfy two properties: First, $g_t$ should be dependent on the accumulated logits of the model up to the time step $t$. Second, $g_t$ should optimize the rate-coded SNN with linearly increasing confidence over time steps as described in Remark 3.2.

### 3.3. Temporal Model Calibration for SNNs

In this section, we propose a Temporal Model Calibration (TMC) method with a new loss function to generate the temporal gradient rescaling factor $g_t$ as discussed above.

**Confidence Regularization.** Inspired by the effective learning-based model calibration methods that add a regularization term to the CE loss (Chen et al., 2024; Pereyra et al., 2017), at each time step, we introduce confidence regularization to the CE loss. The loss function can be expressed as:

$$L_t = L_{\text{CE}}(\mathbf{P}_t, \mathbf{Y}) + \hat{P}_t, \tag{11}$$

$$\hat{P}_t = \text{argmax}(\text{softmax}(\bar{\mathbf{Z}}_t)), \tag{12}$$

$$\bar{\mathbf{Z}}_t = \frac{1}{t} \sum_{i=1}^{t} \mathbf{Z}_i, \tag{13}$$

where $\hat{P}_t$ is the temporal confidence of a rate-coded SNN as defined in Section 3.2.

However, directly using $\hat{P}_t$ as a regularization term is likely to cause suboptimal solutions and could hinder the gradient analysis due to two issues: First, the class index corresponding to the confidence $\hat{P}_t$ may vary during training. While our goal is to optimize $\hat{P}_t$ only when its class index corresponds to the target class. In cases where this is not true, we aim to optimize the probability of the target class to be the highest value in the probability distribution. Second, in Equation 13, $\hat{P}_t$ is generated by the model's temporal outputs up to the current time step $t$. This leads to a complex calculation graph during gradient backpropagation and complicates the formulation of $g_t$.

*Modification 1.* To address the above problems, we replace $\hat{P}_t$ with the output predictive probability of the target class, denoted as $\beta_t$. In addition, we detach the historical logits in Equation 13 from the calculation graph, retaining only the logit at the current time step $t$. The modification can be defined as:

$$\beta_t = \frac{\exp(Q_t^k)}{\sum_j^C \exp(Q_t^j)},$$
$$Q_t = \frac{1}{t} \Big[ \sum_{i=1}^{t-1} \text{SG}(\mathbf{Z}_i) + \mathbf{Z}_t \Big]. \tag{14}$$

where $Q_t$ is the average accumulated logits, $k$ refers to the index of the target class, $C$ is the total number of class, and $\text{SG}(\cdot)$ indicates the stop-gradient operation. We need to highlight that, under the application of $\beta_t$, our loss function's effect is to optimize the probabilities of the target class to serve as the confidence over time steps during training. Thus, in the following content, the optimization objective "confidence" refers to the probability of the target class.

*Modification 2.* The term $\beta_t$ regularizes the logit value of the target class during training and is already able to alleviate overconfidence to some extent. However, we want to explicitly enhance the effect of regularization when $\beta_t$ is close to 1, that is, the model faces a high risk of overconfidence. Thus, we replace $\beta_t$ with $\theta_t$:

$$\theta_t = \frac{\beta_t}{1 - \beta_t} = \frac{\exp(Q_t^k)}{\sum_j^{C, j \neq k} \exp(Q_t^j)}, \tag{15}$$

where $\theta_t$ is highly sensitive to the change of $\beta_t$. When $\beta_t$ approaches 1, even a small change in $\beta_t$ can result in significant increase of $\theta_t$.

**Linear Increasing Confidence Constraint.** Notably, the use of $\text{SG}(\cdot)$ will not hinder our training objective of the linearly increasing confidence over time steps. Specifically, we can prove that, under the guidance of the non-optimizable constant bias, if the logit of the target class $Z_t^k$ at each time step can be optimized to increase monotonically, the accumulated logit value of the target class $\bar{Z}_t^k$ will also increase monotonically. Consequently, the temporal confidence can achieve a linear increase over time steps. To formalize this, we have the following proposition (a detailed proof can be found in Appendix C):

**Proposition 3.4.** *Let $\mathbf{Z}_t^k$ be the logits at a single time step $t$, and $\bar{\mathbf{Z}}_t^k$ be the average logits of the cumulative time step. Given a perfectly calibrated SNN, if for any $t$, $t = 1, ..., T - 1$, $\mathbf{Z}_t^k < \mathbf{Z}_{t+1}^k$, it satisfies $\bar{\mathbf{Z}}_t^k < \bar{\mathbf{Z}}_{t+1}^k$.*

This implies that the effect of $\theta_t$ should decrease as $t$ increases. Given that minor changes in $\beta_t$ can lead to significant differences in $\theta_t$, it is necessary to introduce an

exponential term $_t$ to $\theta_t$ to control its effect. Moreover, we define $\lambda_t = \frac{T-t}{T}$, which linearly decreases with time steps, to ensure that the effect of $\theta_t$ can decrease across time steps. Overall, the final version of the loss function in our method is presented as follows:

$$L_t = L_{\text{CE}}(\boldsymbol{P}_t, \boldsymbol{Y}) + \theta_t^{\lambda_t}. \quad (16)$$

**Gradient Rescaling Factor Analysis.** After designing the loss function of our method, we use it to instantiate the gradient rescaling factor $g_t$ and see if it can match the desired property discussed in Section 3.2.

We first apply the temporal gradient calculation analysis framework in Section 3.1 to our method and derive the expression of the partial derivative of our loss function to the logits $\nabla \boldsymbol{Z}_t^{\text{TMC}}$ as follows:

$$\begin{aligned} \nabla \boldsymbol{Z}_t^{\text{TMC},k} &= P_t^k - Y^k + \frac{1}{t}\lambda_t \theta_t^{\lambda_t}, \\ \nabla \boldsymbol{Z}_t^{\text{TMC},m} &= P_t^m - Y^m - \frac{1}{t}\lambda_t \theta_t^{\lambda_t} \frac{\bar{P}_t^m}{1 - \bar{P}_t^k}. \end{aligned} \quad (17)$$

Here $\bar{P}_t = \text{softmax}(Q_t)$. $k$ and $m$ refer to the index of the target class and the index of the non-target classes, respectively. Then, the temporal logit gradient rescaling vector $g_t$ of our method can be written as

$$\begin{aligned} g_t^k &= \frac{\nabla \boldsymbol{Z}_t^{\text{TCM},k}}{\nabla \boldsymbol{Z}_t^{\text{CE},k}} = 1 - \frac{1}{t}\frac{\lambda_t \theta_t^{\lambda_t}}{1 - P_t^k}, \\ g_t^m &= \frac{\nabla \boldsymbol{Z}_t^{\text{TCM},m}}{\nabla \boldsymbol{Z}_t^{\text{CE},m}} = 1 - \frac{1}{t}\frac{\lambda_t \theta_t^{\lambda_t}}{1 - \bar{P}_t^k}\frac{\bar{P}_t^m}{P_t^m}. \end{aligned} \quad (18)$$

The detailed derivations of Equation 17 and Equation 18 can be found in Appendix D.

In Equation 18, as time step increases, the terms $\frac{1}{t}$ and $\lambda_t$ weaken the effect of $\frac{1}{t}\frac{\lambda_t \theta_t^{\lambda_t}}{1 - P_t^k}$ and $\frac{1}{t}\frac{\lambda_t \theta_t^{\lambda_t}}{1 - \bar{P}_t^k}\frac{\bar{P}_t^m}{P_t^m}$ in $g_t^k$ and $g_t^m$, respectively. This means the logit gradient effect of the CE loss will take over, making the later time steps tend to be more confident. Thus, the linearly increasing temporal confidence of the trained SNNs will align with the monotonically increasing accuracies to satisfy the Definition 3.1. More detailed theoretically analysis of the temporal confidence of a SNN trained with TMC could converge to its accuracy can be found in Appendix E.

We further analyze the diverse distribution of $g_t$ across time steps. From Equation 18, we can infer that the distribution of $g_t$ will be affected by the confidence, that is, $g_t$ has a negative correlation with $P_t^k$ and $\bar{P}_t^k$. Since the temporal confidence monotonically increases with time steps, this will introduce diversity to the distribution of $g_t$ across time steps. We carried out experiments in Section 4.1 to demonstrate the detailed correlation between $g_t$ and confidence, as well as their diversity of $g_t$ over time steps.

## 4. Experiments

We primarily validate our proposed method TMC on the image classification tasks, where static datasets (CIFAR10, CIFAR100, and ImageNet) and neuromorphic datasets (DVSCIFAR10, N-Caltech101, DVS-Gesture, and SL-Animals-DVS) are used. The network architectures in this paper include ResNet-19, VGG-SNN, Meta-SpikeFormer, and Hierarchical SpikingTransformer. More details of the experimental configurations can be found in Appendix A.1. The results on the DVS-Gesture and SL-Animals-DVS are reported in Appendix F.2. Additionally, to further demonstrate the effectiveness of our method, we conduct experiments on the text classification task, where the Quora Question Pair (QQP) dataset and the SpikingBERT architecture (Bal & Sengupta, 2024) are used. The results of this task are reported in Appendix F.1.

### 4.1. Analysis of Temporal Gradient Rescaling Effect

We decompose $g_t$ into the temporal logit gradient rescaling factor on the target class $g_t^k$ and the factor on non-target classes $\bar{g}_t^{-k} = \frac{1}{C-1}\sum_{m=1}^{C,m\neq k} g_t^m$, where $C$ is the total class number. Then, in Figure 1, we visualize the relationship between confidence and $g_t$ as well as the relationship between (1-confidence) and $\bar{g}_t^{-k}$ of 500 samples at the end of training on DVSCIFAR10. There are two important observations: Firstly, at each time step, we can observe that $g_t$ and $\bar{g}_t^{-k}$ show the symmetric distribution pattern. This implies that despite $g_t^k$ and $g_t^m$ having slightly different formulations in Equation (18), the rescaling factor across different classes may have close values with little variation. In such way, the rescaling mechanism will not bring much variation to the direction of temporal gradient. This property is similar to the rescaling factor of some existing model calibration methods, the expectation of which has been proven to be consistent across different classes (Tang et al., 2020). In addition, when the network is under-confident, $g_t^k$ and $\bar{g}_t^{-k}$ will be closer to 1. Similarly, they will likely be pushed toward 0 or become negative numbers when the network is overconfident. Secondly, over time steps, the distributions of confidence and rescaling factors increasingly converge towards 1. This is compatible with what we discussed in Section 3.3 of Gradient Rescaling Factor Analysis. Overall, $g_t$ exhibits diversity distribution in the temporal dimension and at the data instance level. In addition, we provide a evaluation of the training stability of $g_t$ in Appendix F.3.

### 4.2. Evaluation of Model Performance

**Classification and Calibration Performances.** To verify the temporal gradient rescaling effect on model classification and calibration performances, we evaluate the calibration errors and accuracies of our method at each time step on DVSCIFAR10 and compare them with SDT and TET.

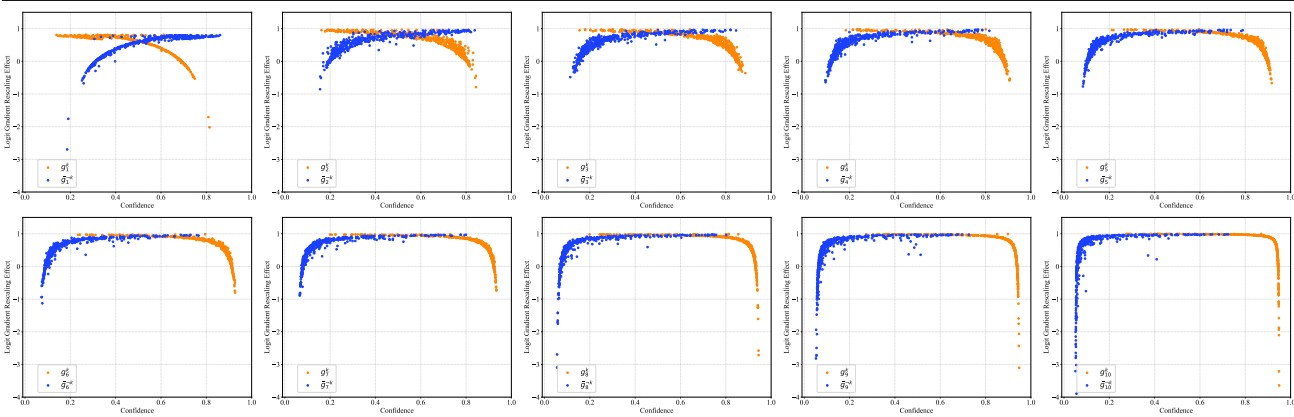

*Figure 1.* Visualization of the temporal relationship between predicted probabilities and gradient rescaling effect.

*Table 1.* Classification and calibration performances for different methods across time steps on DVSCIFAR10.

| Methods | T=1 | | | T=2 | | | T=3 | | | T=4 | | | T=5 | | |
| --- | --- | --- | --- | --- | --- | --- | --- | --- | --- | --- | --- | --- | --- | --- | --- |
| | Acc↑ | ECE↓ | AdaECE↓ | Acc↑ | ECE↓ | AdaECE↓ | Acc↑ | ECE↓ | AdaECE↓ | Acc↑ | ECE↓ | AdaECE↓ | Acc↑ | ECE↓ | AdaECE↓ |
| SDT | 19.24 | 0.64 | 0.69 | 48.42 | 0.36 | 0.42 | 62.06 | 0.27 | 0.24 | 69.24 | 0.22 | 0.20 | 67.28 | 0.21 | 0.15 |
| TET | **65.64** | 0.26 | 0.27 | 73.22 | 0.22 | 0.22 | 77.26 | 0.18 | 0.18 | 78.45 | 0.18 | 0.17 | 79.64 | 0.18 | 0.17 |
| Ours | 63.65 | **0.25** | **0.24** | **75.56** | **0.11** | **0.17** | **78.97** | **0.11** | **0.16** | **81.76** | **0.12** | **0.15** | **81.64** | **0.12** | **0.13** |

| Methods | T=6 | | | T=7 | | | T=8 | | | T=9 | | | T=10 | | |
| --- | --- | --- | --- | --- | --- | --- | --- | --- | --- | --- | --- | --- | --- | --- | --- |
| | Acc↑ | ECE↓ | AdaECE↓ | Acc↑ | ECE↓ | AdaECE↓ | Acc↑ | ECE↓ | AdaECE↓ | Acc↑ | ECE↓ | AdaECE↓ | Acc↑ | ECE↓ | AdaECE↓ |
| SDT | 69.63 | 0.25 | 0.18 | 71.25 | 0.23 | 0.20 | 70.06 | 0.25 | 0.22 | 69.63 | 0.25 | 0.20 | 70.05 | 0.26 | 0.18 |
| TET | 79.22 | 0.18 | 0.17 | 79 67 | 0.17 | 0.17 | 78 45 | 0.17 | 0.17 | 78.04 | 0.17 | 0.17 | 81.22 | 0.18 | 0.16 |
| Ours | **81.91** | **0.11** | **0.13** | **83.37** | **0.08** | **0.08** | **83.08** | **0.06** | **0.06** | **83.87** | **0.04** | **0.05** | **83.94** | **0.08** | **0.05** |

Following previous studies, we use three standard metrics for network calibration performance: Expected Calibration Error (ECE) (Naeini et al., 2015), Adaptive ECE (AdaECE) (Mukhoti et al., 2020), and Classwise-ECE (CECE) (Kull et al., 2019). The specific introduction and calculation formulas for these three metrics can be found in Appendix A.2. We report the main results in Table 1 and the calibration errors in CECE metric are reported in Table A.5 in Appendix F.5. Firstly, compared to SDT and TET, over time steps, our method exhibits very competitive classification accuracies. Compared to TET, our method achieves the highest accuracy improvement of 5.83% at time step 9. Apart from a slight fluctuation at time step 8, overall, the accuracy of our method can increase progressively with the time step, while SDT and TET don't have this property. Secondly, the calibration errors of our method drop significantly in both ECE and AdaECE metrics. In particular, the calibration errors are lower in the later time steps since our proposed method has more advantages in mitigating overconfidence in this phase. In Table A.1, We further show the overall classification and calibration performances by evaluating the average predicted probability across time steps of SNNs. Note that we chose QKformer architecture to evaluate the performance on ImageNet since it is the best-performance architecture. Noticeably, our method achieves the best accuracy and lowest calibration errors on various datasets.

**Temporal Scalability.** The temporal heterogeneity enables SNNs to extract dynamic features and produce temporally extended diverse outputs. This implies that, with increasing time steps, the performance of SNNs can be improved by incorporating diverse temporal outputs. Here, we verify the temporal scalability of SNNs trained with our proposed loss function. Specifically, we first train a ResNet-19 on CIFAR100 with a small time step length of 6 on CIFAR100. Then, we evaluate the trained SNN directly changing the time step length from 2 to 10 without fine-tuning on the test set. Figure 2 displays the results under different time step lengths of our method, SDT and TET. We can see that just under the time step length of 2, our method has surpassed the accuracy of SDT and TET under the time step length of 10. When we increase the test time step length, the accuracy of our method exhibits a more explicit increase, while the accuracy of SDT has a limited increase trend and that of TET remains stable. This implies that the SNNs trained with our method can generate diverse temporal outputs and further introduce significant performance improvement as time step increases.

### 4.3. Ablation Studies

The proposed regularization term consists of two components: the base $\theta_t$ and the exponent $\lambda_t$. In this section, we verify the impact of these two components on the proposed regularization term, respectively.

*Table 2.* Overall classification and calibration performances.

| Dataset | Architecture | Time Step | Method | Accuracy↑ | ECE↓ | AdaECE↓ |
|---|---|---|---|---|---|---|
| **CIFAR100** | **ResNet-19** | 6 | SDT | 71.12 | 0.22 | 0.20 |
| | | | TET | 74.72 | 0.18 | 0.18 |
| | | | Ours | **78.05** | **0.05** | **0.04** |
| **ImageNet** | **QKformer** | 4 | SDT | 85.62 | 0.05 | 0.05 |
| | | | TET | 85.66 | 0.05 | 0.04 |
| | | | Ours | **85.83** | **0.02** | **0.02** |
| **DVSCIFAR10** | VGG-SNN | 10 | SDT | 73.86 | 0.20 | 0.19 |
| | | | TET | 83.17 | 0.16 | 0.14 |
| | | | Ours | **87.63** | **0.12** | **0.11** |
| **N-Caltech101** | VGG-SNN | 10 | SDT | 79.86 | 0.14 | 0.14 |
| | | | TET | 81.72 | 0.11 | 0.10 |
| | | | Ours | **86.03** | 0.12 | 0.10 |

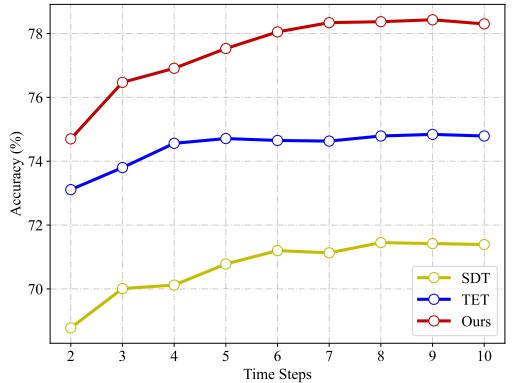

*Figure 2.* Time scalability of ResNet-19 on CIFAR100.

**Effect of $\theta_t$.** To validate the regularization effect introduced by $\theta_t$, we first create a variant of our loss function by replacing $\theta_t$ with $\beta_t$, denoted as Ours*. Then, we compare the classification and calibration performances of the SNNs trained using our loss function and the variant loss function on the DVSCIFAR10 dataset. In Table 3, we report the calibration errors in ECE and AdaECE metrics and test accuracies at each time step on DVSCIFAR10. We can see that the calibration errors of the variant loss function are higher than those of our proposed loss function at the later time steps. In addition, the classification performance at the later time steps is limited. This indicates that $\theta_t$ in our loss function exhibits a more efficient effect for the later time steps to mitigate overconfidence, and further improve the performance of SNNs.

**Effect of $\lambda_t$.** We conduct experiments to evaluate the temporal constraint of $\lambda_t$ on the strength of the proposed regularization term. First, We create a variant of our loss function by removing $\lambda_t$, denoted as Ours†. Then, we compare the classification and calibration performances on DVSCIFAR10 and the results reported in Table 3. Except for the first time step, the calibration errors of the variant loss function are significantly higher than those of our proposed loss function. Moreover, the accuracy does not increase with time step. This means that without the constraint of $\lambda_t$, $\theta_t$ can not have a desirable regularization effect to make temporal confidence increase progressively.

## 4.4. Comparison to Existing Works

In this section, we conduct experiments on static datasets (CIFAR10, CIFAR100, and ImageNet) and neuromorphic datasets (DVSCIFAR10 and N-Caltech101) to evaluate the performance of our method and compare the experimental results with previous works. All of the experiment results are summarized in Table 4. We specify all the training details in Appendix A.1.

**CIFAR10/100.** We apply our method on CIFAR10 and CIFAR100 and report the mean and standard deviation of 3 runs under different random seeds. The results of our method and existing methods on CIFAR100 are shown in Table 4, and the results on CIFAR10 can be found in Table A.4 in Appendix F.4. Our method achieves competitive performance on CIFAR10 and CIFAR100 compared with other methods. The accuracies of our method on these two datasets are slightly lower than TEBN and RMP-Loss since they apply extra membrane potential regularizing operations. Our method outperforms the finite difference surrogate gradient descent method DSpike and spiking neurons parametric method GLIF on CIFAR10 and CIFAR100. compared to TET, TKS, TCL, ETC, and TSSD, our method doesn't achieve the best performance since the superiority of our method can not be fully exhibited on the static datasets.

**ImageNet.** Compared to the convolution-based SNNs, transformer-based SNNs have achieved superior performance on ImageNet. We apply our loss function to the two best-performing transformer-based SNNs: Spike-Driven Transformer V2 and QKFormer. We train Spike-Driven Transformer V2 from scratch. Note that considering the high training cost required to train QKFormer from scratch, we load the trained model of the initial QKformer and fine-tune it using our proposed loss function for 10 epochs. As shown in Table 4, our method further improves the performance of these two architectures by 0.80% and 0.18%, respectively. Overall, our method achieves SOTA performance on ImageNet and exhibits significant accuracy advantages compared to the convolution-based SNNs.

**Neuromorphic Datasets.** DVSCIFAR10 and N-Caltech101 are the most challenging mainstream neuromorphic datasets. We apply our method on these two datasets and report the mean and standard deviation of 3 runs under different random seeds. As shown in Table 4, on DVSCIFAR10, our method achieves an accuracy of 87.63% top-1 accuracy, outperforming the previous SOTA method. On N-Caltech101, the proposed method achieves 88.24% top-1 accuracy. The results show the potential of SNNs in handling neuromorphic data. Compared to static datasets, neuromorphic datasets exhibit richer spatiotemporal components through their interaction with both spatial and temporal information. TMC leverages temporal heterogeneous training, thereby capturing a richer set of task-relevant spatiotempo-

Table 3. Ablation study on DVSCIFAR10.

| Methods | T=1 | | | T=2 | | | T=3 | | | T=4 | | | T=5 | | |
|---|---|---|---|---|---|---|---|---|---|---|---|---|---|---|---|
| | ACC↑ | ECE↓ | AdaECE↓ | ACC↑ | ECE↓ | AdaECE↓ | ACC↑ | ECE↓ | AdaECE↓ | ACC↑ | ECE↓ | AdaECE↓ | ACC↑ | ECE↓ | AdaECE↓ |
| Ours* | 65.00 | 0.25 | 0.25 | 78.56 | 0.17 | 0.16 | 78.09 | 0.17 | 0.17 | 79.26 | 0.16 | 0.16 | 80.08 | 0.16 | 0.16 |
| Ours† | 64.40 | 0.26 | 0.26 | 76.20 | 0.22 | 0.23 | 77.05 | 0.20 | 0.22 | 78.80 | 0.21 | 0.20 | 79.80 | 0.20 | 0.21 |
| Ours | 63.65 | 0.25 | **0.24** | 75.56 | **0.11** | **0.17** | **78.97** | **0.11** | **0.16** | **81.76** | **0.12** | **0.15** | 81.64 | **0.12** | **0.13** |

| Methods | T=6 | | | T=7 | | | T=8 | | | T=9 | | | T=10 | | |
|---|---|---|---|---|---|---|---|---|---|---|---|---|---|---|---|
| | ACC↑ | ECE↓ | AdaECE↓ | ACC↑ | ECE↓ | AdaECE↓ | ACC↑ | ECE↓ | AdaECE↓ | ACC↑ | ECE↓ | AdaECE↓ | ACC↑ | ECE↓ | AdaECE↓ |
| Ours* | 80.85 | 0.16 | 0.17 | 81.08 | 0.16 | 0.16 | 81.38 | 0.17 | 0.16 | 81.35 | 0.17 | 0.16 | 81.82 | 0.16 | 0.16 |
| Ours† | 81.20 | 0.17 | 0.17 | 80.40 | 0.17 | 0.16 | 81.02 | 0.17 | 0.17 | 80.80 | 0.18 | 0.18 | 81.32 | 0.17 | 0.16 |
| Ours | **81.91** | **0.11** | **0.13** | **83.37** | **0.08** | **0.08** | **83.08** | **0.06** | **0.06** | **83.87** | **0.04** | **0.05** | **83.94** | **0.08** | **0.05** |

Table 4. Performance comparison with state-of-the-art methods.

| Dataset | Model | Architecture | Time Step | Accuracy |
|---|---|---|---|---|
| CIFAR100 | DSpike(Li et al., 2021) | ResNet-18 | 6 | 74.24 |
| | GLIF(Yao et al., 2022) | ResNet-19 | 6 | 77.35 |
| | TEBN(Duan et al., 2022) | ResNet-19 | 6 | 78.76 |
| | RMP-Loss(Guo et al., 2023) | ResNet-19 | 6 | 78.98 |
| | TET(Deng et al.) | ResNet-19 | 6 | 74.72 |
| | TKS(Dong et al., 2024) | ResNet-19 | 4 | 76.20 |
| | TCL(Qiu et al., 2024) | ResNet-19 | 4 | 79.73 |
| | ETC(Zhao et al., 2025) | ResNet-19 | 4 | 79.47 |
| | TSSD(Zuo et al., 2024) | VGG-9 | 2 | 74.69 |
| | **Ours** | ResNet-19 | 6 | **78.05±0.18** |
| | | | 4 | **77.52±0.13** |
| | | | 2 | **76.35±0.15** |
| ImageNet | DSpike(Li et al., 2021) | VGG-16 | 5 | 71.24 |
| | GLIF(Yao et al., 2022) | ResNet-34 | 6 | 69.09 |
| | TEBN(Duan et al., 2022) | SEW ResNet-34 | 4 | 68.28 |
| | RMP-Loss(Guo et al., 2023) | ResNet-34 | 4 | 65.17 |
| | TET(Deng et al.) | Spiking-ResNet-34 | 6 | 64.79 |
| | TKS(Dong et al., 2024) | SEW-ResNet-34 | 4 | 69.60 |
| | ETC(Zhao et al., 2025) | Spiking-ResNet-34 | 6 | 69.64 |
| | TSSD(Zuo et al., 2024) | ResNet-34 | 2 | 66.13 |
| | Spike-Driven Transformer V2(Yao et al.) | Meta-SpikeFormer | 4 | 80.00 |
| | QKFormer(Zhou et al., 2024) | Hierarchical SpikingTransformer | 4 | 85.65 |
| | **Ours** | Meta-SpikeFormer | 4 | **80.80** |
| | | Hierarchical SpikingTransformer | 4 | **85.83** |
| DVSCIFAR10 | DSpike(Li et al., 2021) | ResNet-18 | 10 | 75.40 |
| | GLIF(Yao et al., 2022) | Wide 7B Net | 16 | 78.10 |
| | TEBN(Duan et al., 2022) | VGG-SNN | 10 | 84.90 |
| | RMP-Loss(Guo et al., 2023) | ResNet-19 | 10 | 76.20 |
| | TET(Deng et al.) | VGG-SNN | 10 | 83.17 |
| | TKS(Dong et al., 2024) | VGG-SNN | 10 | 85.30 |
| | TCL(Qiu et al., 2024) | VGG-SNN | 4 | 79.10 |
| | ETC(Zhao et al., 2025) | VGG-SNN | 10 | 85.95 |
| | TSSD(Zuo et al., 2024) | VGG-9 | 16 | 84.37 |
| | **Ours** | VGG-SNN | 10 | **87.63±0.15** |
| N-Caltech101 | DART(Ramesh et al., 2019) | N/A | N/A | 66.80 |
| | TCJA-SNN(Zhu et al., 2024) | VGG-SNN | 14 | 78.50 |
| | STCA-SNN(Wu et al., 2023) | N/A | 14 | 80.90 |
| | TET(Deng et al.) | VGG-SNN | 10 | 81.72 |
| | ETC(Zhao et al., 2025) | VGG-SNN | 10 | 85.53 |
| | TKS(Dong et al., 2024) | VGG-SNN | 10 | 84.10 |
| | **Ours** | VGG-SNN | 10 | **86.03±0.13** |
| | | VGG-SNN | 16 | **88.24±0.10** |

ral dynamics features. This enables TMC to outperform existing methods that rely on temporal homogeneous training.

## 5. Conclusion

In this work, we analyze the temporal logit gradient in existing direct training methods and conclude that the insufficient diversity of the logit gradient in the temporal dimension will limit the temporal heterogeneity and the performance of SNNs. Inspired by model calibration methods, we propose TMC, which can be seen as a temporal gradient rescaling mechanism to generate diverse logit gradients in the temporal dimension. Experimental results show that our method can improve the temporal heterogeneity and performance of SNNs. In particular, our method achieves state-of-the-art accuracy on ImageNet, DVSCIFAR10, and N-Caltech101.

## Acknowledgements

This work was supported in part by the National Key Research and Development Program of China (No. 2020AAA0109002), in part by the National Natural Science Foundation of China (62376247, 62334014, and 62436008), and in part by the grant from Key R&D Program of Zhejiang (2022C01048).

## Impact Statement

This paper presents work whose goal is to advance the field of Machine Learning. There are many potential societal consequences of our work, none which we feel must be specifically highlighted here.

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

# A. Details of Experiments

## A.1. Datasets and Training Details

**CIFAR10/100.** CIFAR10/10 (Krizhevsky et al., 2009) contains 10/100 classes and consists of 50k training images and 10k testing images with the size of $32 \times 32$. The original random horizontal flip and crop are applied to the training image augmentation. We train ResNet-19 architecture on these two datasets under time step lengths of 2, 4, and 6 for 300 epochs, respectively. We use an Adam optimizer with a learning rate of 0.01 and apply cosine decay, gradually reducing the learning rate to 0.

**ImageNet.** ImageNet (Deng et al., 2009) is the challenging image recognition dataset with 1.28 million training images and 50k test images from 1000 object classes. We crop the images to 224×224 and use the standard augmentation for the training data. The detailed experimental settings are the same as Spike-Driven Transformer V2 (Yao et al.) and QKformer (Zhou et al., 2024).

**DVSCIFAR10.** DVSCIFAR10 (Li et al., 2017) is converted from CIFAR10. It has 10k images with a size of 128×128. We apply a $9:1$ train-test split (i.e. 9k training images and 1k test images) to DVSCIFAR10. In our training, we integrate the event data into 10 frames and resize the resolution to $48 \times 48$. Random horizontal flip and random roll within 5 pixels are taken as augmentation (Li et al., 2022). We train VGG-SNN architecture on DVSCIFAR10 under time step lengths of 10 for 300 epochs and use an Adam optimizer with a learning rate of 0.01 and cosine decay to 0.

**N-Caltech101.** N-Caltech101 (Orchard et al., 2015) is a neuromorphic version of the Caltech101 dataset, encompassing 101 categories. We train VGG-SNN architecture on DVSCIFAR10 under time step lengths of 10 and 14 for 300 epochs, respectively. In addition, we use an Adam optimizer with a learning rate of 0.01 and cosine decay to 0.

**DVS-Gesture.** DVS-Gesture (Amir et al., 2017) is a neuromorphic dataset for gesture recognition. DVS-Gesture contains a total of 11 event stream samples of gestures, 1176 for training and 288 for testing, with a spatial size of 128×128 for each sample. For DVS-Gesture data, the event stream is integrated into frames in 30ms units and downsampled to $32 \times 32$. We train VGG-SNN architecture on DVS-Gesture under time step lengths of 10 for 300 epochs. In addition, we use an Adam optimizer with a learning rate of 0.01 and cosine decay to 0.

**SL-Animals-DVS.** SL-Animals-DVS (Vasudevan et al., 2022) is a event-based Spanish sign language dataset. It is composed of more than 1100 samples of 59 subjects performing 19 signs in isolation corresponding to animals. We train VGG-SNN architecture on SL-Animals-DVS under time step lengths of 16 for 300 epochs and use an Adam optimizer with a learning rate of 0.01 and cosine decay to 0.

**Quora Question Pair (QQP).** The QQP dataset (Sharma et al., 2019) consists of a training set of 404,290 question pairs, and a test set of 2,345,795 question pairs and is provided as part of a Kaggle competition. The detailed experimental settings are the same as SpikingBERT (Bal & Sengupta, 2024).

## A.2. Calibration Metrics

**ECE.** The ECE of SNN at time step $t$ can be defined as:

$$\text{ECE}_t = \mathbb{E}_{\hat{p_t}}[|\mathbb{P}(\hat{y_t} = y_t|\hat{p_t}) - \hat{p_t}|]. \tag{A.1}$$

Since we only have finite samples, an approximate estimation is used to calculate $\text{ECE}_t$ given a finite sample size of $N$. Specifically, we group the probability predictions at time step $t$ into $M$ equispaced bins. Let $B_{m,t}$ denote the set of samples with predicted confidence belonging to the $m_{th}$ bin, where the interval is $[\frac{i-1}{M}, \frac{i}{M}]$. Then, $\mathbb{P}(\hat{y_t} = y_t|\hat{p_t})$ in Equation (A.1) can be denoted as the accuracy of $B_{m,t}$: $A_{m,t} = \frac{1}{|B_{m,t}|} \sum_{i \in B_{m,t}} \mathbb{1}(\hat{y_t^i} = y_t^i)$, where $\mathbb{1}$ is the indicator function. Similarly, $\hat{p_t}$ in Equation (A.1) can be represented as the mean confidence of $B_{m,t}$, that is the average confidence of all samples in the bin: $C_{m,t} = \frac{1}{|B_{m,t}|} \sum_{i \in B_{m,t}} \hat{p_t^i}$. Then, $\text{ECE}_t$ can be approximated as a weighted average of the absolute difference between the accuracy and confidence of each bin:

$$\text{ECE}_t = \sum_{m=1}^{M} \frac{|B_{m,t}|}{N} |A_{m,t} - C_{m,t}|. \tag{A.2}$$

**AdaECE.** One disadvantage of ECE is the uniform bin width. For a trained model, most of the samples lie within the highest confidence bins, and hence these bins dominate the value of the ECE. We thus also consider another metric, AdaECE, for

which bin sizes are calculated so as to evenly distribute samples between bins (similar to the adaptive binning procedure in (Nguyen & O'Connor, 2015)). The $\text{AdaECE}_t$ of SNN at time step $t$ can be calculated as:

$$\text{AdaECE}_t = \sum_{m=1}^{M} \frac{|B_{m,t}|}{N} |A_{m,t} - C_{m,t}| \quad \text{s.t.} \forall i, j \cdot |B_{m,t}| = |B_{j,t}|. \tag{A.3}$$

**CECE.** The ECE metric only considers the probability of the predicted class, without considering the other scores in the softmax distribution. A stronger definition of calibration would require the probabilities of all the classes in the softmax distribution to be calibrated. This can be achieved with a simple classwise extension of the ECE metric. The $\text{CECE}_t$ of SNN at time step $t$ can be calculated as:

$$\text{CECE}_t = \frac{1}{K} \sum_{m=1}^{M} \sum_{j=1}^{K} \frac{|B_{m,t}^j|}{N} |A_{m,t}^j - C_{m,t}^j|, \tag{A.4}$$

where $K$ is the number of classes, $B_{m,t}^j$ denotes the set of samples from the $j$-th class in the $m$-th bin at time step $t$, $A_{m,t}^j = \frac{1}{|B_{m,t}^j|} \sum_{k \in B_{m,t}^j} \mathbb{1}(j = y_t^k)$ and $C_{m,t}^j = \frac{1}{|B_{m,t}^j|} \sum_{k \in B_{m,t}^j} p_t^{k,j}$.

Note that we set the number of bins for ECE and CECE as 15 in our experiments.

## B. Detailed Analysis of the Temporal Heterogeneity in SNNs

### B.1. Neuronal Behavior Definition

The behavior of a spiking neuron over continuous time can be formulated as follows:

$$\tau \frac{du_t}{dt} = -(u_t - u_{reset}) + I_t, \quad u_t < V_{th}, \tag{A.5}$$

$$o_t = \sum_{t_f} \delta(t - t_f), \tag{A.6}$$

where $u_t$ is the membrane potential, $\tau$ is the time content, $I_t$ is the input current, and $V_{th}$ is the threshold. $\delta(\cdot)$ is the Dirac delta function. The spike train $o_t$ is generated when $u_t$ reaches $V_{th}$ at time $t_f$, and $u_t$ is reset to the resting potential $u_{reset}$.

### B.2. Neuronal Temporal Heterogeneity Analysis

Firstly, the change of $u_t$ is quantified by $\frac{du_t}{dt}$, which is determined by two components:

1) Decay of Membrane Potential $-(u_t - u_{reset})$. This term indicates that the membrane potential naturally decays towards the resting potential $u_{reset}$. If $u_t > u_{reset}$, this term is negative, indicating a decrease in membrane potential. If $u_t < u_{reset}$, this term is positive, indicating an increase in membrane potential.

2) Input Current $I_t$. It represents the effect of the external current on the membrane potential. If $I_t$ is positive, it drives the membrane potential up. If $I_t$ is negative, it pushes the membrane potential down.

Secondly, we investigate the temporal dynamic changes of $\frac{du_t}{dt}$:

1) When $\frac{du_t}{dt} > 0$, it means the membrane potential $u_t$ is increasing. This could be due to an input current $I_t$ that is sufficiently large to overcome the natural decay of the membrane potential, or because the effect of the natural decay is relatively small at the current moment.

2) When $\frac{du_t}{dt} < 0$, the membrane potential $u_t$ is decreasing. This could be because the input current $I_t$ is small or negative, not enough to counteract the decay of the membrane potential, or the effect of the natural decay is relatively large at the current moment.

3) Note that $\frac{du_t}{dt} = 0$ is highly unlikely to occur unless specific conditions are met.

Thus, $|\frac{du_t}{dt}| > 0$ indicates that the membrane potential is continuously changing, either increasing or decreasing. This change is a direct reflection of the neuron's response to temporal dependency and input signals and is key evidence of temporal heterogeneity.

## C. Proof of Proposition 3.4

*Proof.* The cumulative logits of the target class, denoted as $\bar{\mathbf{Z}}_t^k$ can be written as $\frac{1}{t}\sum_{i=1}^t \mathbf{Z}_i^k$. Thus we can get

$$
\begin{aligned}
&\bar{\mathbf{Z}}_t^k < \bar{\mathbf{Z}}_{t+1}^k \\
\Rightarrow \quad & \frac{1}{t}\sum_{i=1}^t \mathbf{Z}_i^k < \frac{1}{t+1}\sum_{i=1}^{t+1} \mathbf{Z}_i^k \\
\Rightarrow \quad & \frac{t+1}{t}\sum_{i=1}^t \mathbf{Z}_i^k < \sum_{i=1}^{t+1} \mathbf{Z}_i^k \\
\Rightarrow \quad & \frac{t+1}{t}\sum_{i=1}^t \mathbf{Z}_i^k - \sum_{i=1}^t \mathbf{Z}_i^k < \mathbf{Z}_{t+1}^k \\
\Rightarrow \quad & \frac{1}{t}\sum_{i=1}^t \mathbf{Z}_i^k < \mathbf{Z}_{t+1}^k.
\end{aligned}
\tag{A.7}
$$

By subtracting $\mathbf{Z}_t^k$ on both sides of Equation A.7, we get

$$
\begin{aligned}
& \frac{1}{t}\sum_{i=1}^t \mathbf{Z}_i^k - \mathbf{Z}_t^k < \mathbf{Z}_{t+1}^k - \mathbf{Z}_t^k \\
\Rightarrow \quad & \frac{1}{t}\sum_{i=1}^t (\mathbf{Z}_i^k - \mathbf{Z}_t^k) < \mathbf{Z}_{t+1}^k - \mathbf{Z}_t^k.
\end{aligned}
\tag{A.8}
$$

If for any $t$, $t = 1, ..., T-1$, $\mathbf{Z}_t^k < \mathbf{Z}_{t+1}^k$, it holds that

$$
\frac{1}{t}\sum_{i=1}^t (\mathbf{Z}_i^k - \mathbf{Z}_t^k) \leq 0 < \mathbf{Z}_{t+1}^k - \mathbf{Z}_t^k.
\tag{A.9}
$$

Hence it satisfies $\bar{\mathbf{Z}}_t^k < \bar{\mathbf{Z}}_{t+1}^k$. $\qquad\square$

## D. Derivations in Section 3.3

### D.1. Derivation of Equation 17

The loss function of our TMC method is:

$$
L_t^{\text{TCM}} = L_{\text{CE}}(\boldsymbol{o}_t, \boldsymbol{Y}) + \theta_t^{\lambda_t}.
\tag{A.10}
$$

According to the chain rule, the partial derivative of this loss function with respect to the logit vector at time step $t$ is:

$$
\begin{aligned}
\nabla \mathbf{Z}_t^{\text{TCM},i} &= \frac{\partial L_{\text{CE}}}{\partial \mathbf{Z}_t^i} + \frac{\partial \theta_t^{\lambda_t}}{\partial \mathbf{Z}_t^i} \\
&= \frac{\partial L_{\text{CE}}}{\partial \mathbf{Z}_t^i} + \frac{\partial \theta_t^{\lambda_t}}{\partial \theta_t}\frac{\partial \theta_t}{\partial \bar{P}_t^k}\frac{\partial \bar{P}_t^k}{\partial Q_t^i}\frac{\partial Q_t^i}{\partial \mathbf{Z}_t^i} + \frac{\partial \theta_t^{\lambda_t}}{\partial \theta_t}\frac{\partial \theta_t}{\partial \bar{P}_t^m}\frac{\partial \bar{P}_t^m}{\partial Q_t^i}\frac{\partial Q_t^i}{\partial \mathbf{Z}_t^i},
\end{aligned}
\tag{A.11}
$$

where superscript $i$ is used to separate $\nabla \mathbf{Z}_t^{\text{TCM}}$ by different classes. Superscript $k$ and $m$ refer to the index of the target class and the index of non-target classes, respectively. For the element of the target class, we have:

$$\frac{\partial L_{\text{CE}}}{\partial \mathbf{Z}_t^k} = P_t^k - Y^k,$$

$$\frac{\partial \theta_t^{\lambda_t}}{\partial \theta_t} = \lambda_t \theta_t^{\lambda_t - 1},$$

$$\frac{\partial \theta_t}{\partial \bar{P}_t^k} = \frac{1}{1 - \bar{P}_t^k},$$

$$\frac{\partial \theta_t}{\partial \bar{P}_t^m} = -\frac{\bar{P}_t^k}{(1 - \bar{P}_t^k)^2}, \tag{A.12}$$

$$\frac{\partial \bar{P}_t^k}{\partial Q_t^k} = \frac{exp(Q_t^k)}{\sum_j^C exp(Q_t^j)} - \frac{exp(Q_t^k)^2}{\left[\sum_j^C exp(Q_t^j)\right]^2} = \bar{P}_t^k - (\bar{P}_t^k)^2,$$

$$\frac{\partial \bar{P}_t^m}{\partial Q_t^k} = -\frac{exp(Q_t^k) \cdot \sum_j^{C, j \neq k} exp(Q_t^j)}{\left[\sum_j^C exp(Q_t^j)\right]^2} = -\bar{P}_t^k \cdot (1 - \bar{P}_t^k),$$

$$\frac{\partial Q_t^k}{\partial \mathbf{Z}_t^k} = \frac{1}{t}.$$

In Equation A.12, $C$ is the total number of classes. By substituting Equation A.12 into Equation A.11, we have:

$$\nabla \mathbf{Z}_t^{\text{TMC, k}} = P_t^k - Y^k + \lambda_t \theta_t^{\lambda_t - 1} \cdot \frac{1}{1 - \bar{P}_t^k} \cdot [\bar{P}_t^k - (\bar{P}_t^k)^2] \cdot \frac{1}{t} + \lambda_t \theta_t^{\lambda_t - 1} \cdot -\frac{\bar{P}_t^k}{(1 - \bar{P}_t^k)^2} \cdot -\bar{P}_t^k \cdot (1 - \bar{P}_t^k) \cdot \frac{1}{t}$$

$$= P_t^k - Y^k + \lambda_t \theta_t^{\lambda_t - 1} \cdot \frac{\bar{P}_t^k}{1 - \bar{P}_t^k} \cdot (1 - \bar{P}_t^k) \cdot \frac{1}{t} + \lambda_t \theta_t^{\lambda_t - 1} \cdot \frac{\bar{P}_t^k}{1 - \bar{P}_t^k} \cdot \bar{P}_t^k \cdot \frac{1}{t} \tag{A.13}$$

$$= P_t^k - Y^k + \lambda_t \theta_t^{\lambda_t} \cdot \frac{1}{t}$$

Finally, we have:

$$\nabla \mathbf{Z}_t^{\text{TMC},k} = P_t^k - Y^k + \frac{1}{t} \lambda_t \theta_t^{\lambda_t}. \tag{A.14}$$

In the same way, we can have:

$$\nabla \mathbf{Z}_t^{\text{TMC},m} = P_t^m - Y^m - \frac{1}{t} \lambda_t \theta_t^{\lambda_t} \frac{\bar{P}_t^m}{1 - \bar{P}_t^k}. \tag{A.15}$$

### D.2. Derivation of Equation 18

The gradient rescaling vector $g_t$ of our method is defined as:

$$\nabla \mathbf{Z}_t^{\text{TMC}} \oslash \nabla \mathbf{Z}_t^{\text{TET}} = g_t. \tag{A.16}$$

Since $\nabla \mathbf{Z}_t^{\text{TMC}}$ has different expressions for the element of the target class and the other elements, $g_t$ should also be derived separately for those two categories. We denote $g_t^k$ as the gradient rescaling factor of the target class, and $g_t^m$ as the gradient rescaling factor of all the other classes.

For the target class, we have:

$$g_t^k = \frac{\nabla \mathbf{Z}_t^{\text{TMC},k}}{\nabla \mathbf{Z}_t^{\text{TET},k}} = \frac{P_t^k - Y^k + \frac{1}{t} \lambda_t \theta_t^{\lambda_t}}{P_t^k - Y^k} = 1 + \frac{\frac{1}{t} \lambda_t \theta_t^{\lambda_t}}{P_t^k - Y^k}. \tag{A.17}$$

The denominator $P_t^k - Y^k$ in Equation A.17 can also be written as $-(1 - P_t^k)$, so Equation A.17 can be re-written as:

$$g_t^k = \frac{\nabla \mathbf{Z}_t^{\text{TMC},k}}{\nabla \mathbf{Z}_t^{\text{TET},k}} = 1 - \frac{1}{t} \frac{\lambda_t \theta_t^{\lambda_t}}{1 - P_t^k}. \tag{A.18}$$

Similarly, for the non-target classes $g_t^m$, we have:

$$g_t^m = \frac{\nabla \mathbf{Z}_t^{\text{TMC},m}}{\nabla \mathbf{Z}_t^{\text{TET},m}} = \frac{P_t^m - Y^m - \frac{1}{t}\lambda_t\theta_t^{\lambda_t}\frac{\bar{P}_t^m}{1-\bar{P}_t^k}}{P_t^m - Y^m} = 1 - \frac{1}{t}\frac{\lambda_t\theta_t^{\lambda_t}}{1-\bar{P}_t^k}\frac{\bar{P}_t^m}{P_t^m}, \tag{A.19}$$

because $Y^m = 0$.

## E. Convergence Analysis of TMC

### E.1. Temporally Perfectly Clibrated SNN

Definition 3.1 requires a rate-coding SNN to satisfy the following two properties:

1) Property 1: $\hat{P}_t = \mathbb{P}(\hat{y} = y | \hat{P}_t), t \in \{1, 2, \ldots, T\}$.

2) Property 2: $\hat{P}_t < \hat{P_{t+1}}$.

### E.2. Optimization Objective

Convert the realization of these two properties into the optimization objective of TMC:

1) Note: Since the predicted outputs of a trained SNN typically have the highest probability for the target class across time steps, $\hat{P}_t$ can be expressed as $\hat{P}_t = \bar{P_t^k}$. Here, $\bar{P}_t^k$ is the probability of the target class $k$ in the distribution of softmax($\bar{\mathbf{Z}}_t$).

2) Objective 1: During training, the realization of Property 1 can be converted to optimize $|\hat{P}_t - \mathbb{P}(\hat{y} = y|\hat{P}_t)| < \epsilon$. This can be achieved by introducing a confidence regularization term, $\theta_t$, to penalize the under-confidence issue $\hat{P}_t < \mathbb{P}(\hat{y} = y|\hat{P}_t)$ and, especially, the over-confidence issue $\hat{P}_t > \mathbb{P}(\hat{y} = y|\hat{P}_t)$ with high sensitivity.

3) Objective 2: During training, the realization of Property 2, $\hat{P}_t < \hat{P_{t+1}}$, can be converted to $\hat{P}_t < \hat{P_{t+1}}$. This can be achieved by introducing a linearly decreasing exponent, $\lambda_t$, into $\theta_t$ to optimize $Z_t^k < Z_{t+1}^k$, as described in Proposition 3.4.

### E.3. Convergence Analysis

With the loss function of TMC, the rescaling factor for the target class $k$ is generated to optimize confidence. Specifically:

$$g_t^k = \frac{\nabla \mathbf{Z}_t^{\text{TCM},k}}{\nabla \mathbf{Z}_t^{\text{CE},k}} = 1 - f(t) * h(t), \quad f(t) = \frac{\lambda_t\theta_t^{\lambda_t}}{t}, \quad h(t) = \frac{1}{1-P_t^k}. \tag{A.20}$$

Here, $f(t)$ decreases with time steps.

1) At the initial training phase, $h(t)$ follows a random uniform distribution, and $g_t^k$ increases with time steps to optimize $Z_t^k < Z_{t+1}^k$, thereby meeting Objective 2.

2) During training, at time step $t$, if $Z_t^k$ is high, the probability of the target class $k$ in the distribution of softmax($\mathbf{Z}_t$), denoted as $P_t^k$, may lead to overconfidence. In this situation, $h(t)$ increases, causing $g_t^k$ to decrease potentially even to a negative value, to penalize the overconfidence issue. Conversely, underconfidence occurs when $Z_t^k$ is low, and $g_t^k$ increases to address this issue, thereby meeting Objective 1.

3) At the end of the training, $g_t^k$ for different samples will converge to an interval. We have visualized the distribution of $g_t^k$ values for 500 samples of a trained SNN in Figure 1. Notably, for most samples with reasonable confidence, their $\boldsymbol{g}_t$ values are centered within an interval that shifts closer to 1 over time. This indicates the achievement of Objective 2. It can be seen that across time step, some samples' $g_t^k$ values are close to 1 or negative numbers. This is consistent with Objective 1, which aims to penalize particularly underconfident and overconfident samples. Overall, TMC has the effect of realizing these two optimization objectives and generates temporally calibrated SNNs.

## F. Experimental Results

### F.1. Performance Comparison on QQP

We apply our method in the direct training phase of SpikingBERT (Bal & Sengupta, 2024) model proposed in () on the widely used dataset QQP. We compare the classification performance of TMC with SDT (as used in the original paper) and

*Table A.1.* Performance comparison for different methods on QQP.

| Dataset | Architecture | Time Step | Method | Accuracy |
|---------|--------------|-----------|--------|----------|
| **QQP** | **SpikingBERT** | 80 | SDT
TET
Ours | 71.12
74.72
**78.05** |

TET. TMC achieves a higher accuracy of 87.86% than SDT(86.82%) and TET(87.03%).

### F.2. Performance Comparison on Neuromorphic Datasets

We train VGGSNN with T=10 on DVS-Gesture and compare the performance of TMC with current works in Table A.2. TMC achieves SOTA performance with an accuracy of 99.12%.

*Table A.2.* Performance comparison with state-of-the-art methods on DVS-Gesture.

| Dataset | Model | Architecture | Time Step | Accuracy |
|---------|-------|--------------|-----------|----------|
| **DVS-Gesture** | STBP-tdBN(Wu et al., 2018) | ResNet-17 | 40 | 96.87 |
| | PLIF(Fang et al., 2021b) | 5Conv, 2FC | 20 | 97.57 |
| | SEW ResNet(Fang et al., 2021a) | 7B-Net | 16 | 97.92 |
| | TA-SNN(Yao et al., 2021) | TA-SNN | 20 | 98.61 |
| | TCJA(Zhu et al., 2024) | 5Conv, 2FC | 20 | 99.00 |
| | STSC(Yu et al., 2022) | 5Conv, 2FC | 20 | 98.96 |
| | SLT-TET(Anumasa et al., 2024) | VGG-SNN | 10 | 98.43 |
| | Spikeformer(Li et al., 2024c) | Spikeformer-7/5 × 1 × 3 | 16 | 98.96 |
| | **Ours** | VGG-SNN | 10 | **99.12** |

We train VGGSNN with T=16 using TMC on SL-Animals-DVS and compare the performance with SDT and TET in Table A.3. TMC achieves a higher accuracy of 70.05% against SDT (66.75%) and TET (68.34%), demonstrating the superiority of TMC.

*Table A.3.* Performance comparison for different methods on SL-Animals-DVS.

| Dataset | Architecture | Time Step | Method | Accuracy |
|---------|--------------|-----------|--------|----------|
| **SL-Animals-DVS** | **VGG-SNN** | 16 | SDT
TET
Ours | 66.75
68.34
**70.05** |

### F.3. Training Stability Evaluation of TMC

We compare the trends of test loss variation during the training process among SDT, TET, and TMC on DVSCIFAR10 in Figure A1. The results demonstrate TMC's stable training, driving loss to minimal values, unlike SDT and TET, which suffer from overconfidence-induced oscillations.

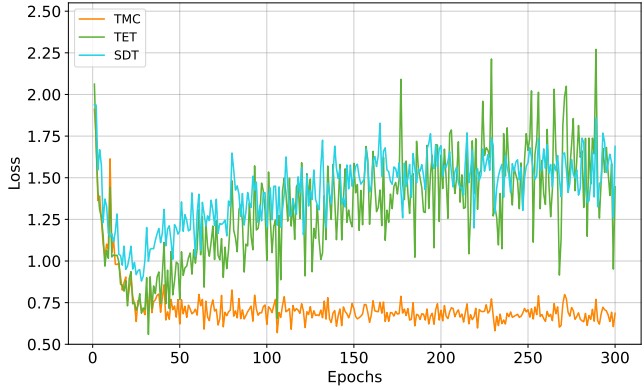

*Figure A1.* Training Stability Evaluation on DVSCIFAR10.

## F.4. Performance Comparison on CIFAR10

*Table A.4.* Performance comparison with state-of-the-art methods on CIFAR10.

| Dataset | Model | Architecture | Time Step | Accuracy |
|---------|-------|--------------|-----------|----------|
| **CIFAR10** | DSpike(Li et al., 2021) | ResNet-18 | 6 | 94.25 |
| | GLIF(Yao et al., 2022) | ResNet-19 | 6 | 95.03 |
| | TEBN(Duan et al., 2022) | ResNet-19 | 6 | 95.60 |
| | RMP-Loss(Guo et al., 2023) | ResNet-19 | 6 | 96.10 |
| | TET(Deng et al.) | ResNet-19 | 6 | 94.50 |
| | TKS(Dong et al., 2024) | ResNet-19 | 4 | 95.30 |
| | TCL(Qiu et al., 2024) | ResNet-19 | 4 | 95.03 |
| | ETC(Zhao et al., 2025) | ResNet-19 | 4 | 95.87 |
| | TSSD(Zuo et al., 2024) | VGG-9 | 2 | 94.41 |
| | **Ours** | ResNet-19 | 6 | **95.23±0.12** |
| | | | 4 | **95.08±0.10** |
| | | | 2 | **94.87±0.08** |

## F.5. Calibration Error in CEC Metric on DVSCIFAR10

*Table A.5.* Calibration error in CECE metric for different methods across time steps on DVSCIFAR10.

| Methods | T=1 | T=2 | T=3 | T=4 | T=5 | T=6 | T=7 | T=8 | T=9 | T=10 |
|---------|-----|-----|-----|-----|-----|-----|-----|-----|-----|------|
| SDT | 0.13 | 0.08 | 0.07 | 0.06 | 0.05 | 0.05 | 0.05 | 0.05 | 0.05 | 0.05 |
| TET | 0.07 | 0.05 | 0.05 | 0.04 | 0.04 | 0.04 | 0.04 | 0.04 | 0.04 | 0.04 |
| Ours | **0.06** | 0.05 | 0.05 | 0.04 | 0.04 | 0.04 | **0.03** | **0.03** | **0.02** | **0.03** |

Compared to SDT and TET, our method achieves lower calibration errors. In particular, the calibration errors of our method drop significantly in the later time steps. This observation further demonstrates that our method can effectively alleviate the overfitting issue.

