# OpenReview forum: "Training High Performance Spiking Neural Network  by Temporal Model Calibration"
_ICML.cc/2025/Conference — ICML 2025 poster_

### Official Review · Reviewer_g8s8 · 2025-03-04

**Overall Recommendation:** 3

**Summary:**

This paper systematically summarizes previous logit gradient calculation schemes, including SDT and TET, then proposes a new temporal gradient rescaling method to enhance the learning capability of SNNs.

**Claims And Evidence:**

Yes

**Essential References Not Discussed:**

Not found yet

**Experimental Designs Or Analyses:**

I have checked experimental designs mentioned in Section 4.

**Methods And Evaluation Criteria:**

I tend to think the proposed method can provide a new perspective for the SNN community to a certain extent.

**Other Comments Or Suggestions:**

This work can be seen as a further exploration on TET [1], which proposes a new loss function calculation scheme around the gradient allocation problem at different time-steps.

[1] Deng, S., Li, Y., Zhang, S., and Gu, S. Temporal efficient training of spiking neural network via gradient reweighting. ICLR 2022.

**Other Strengths And Weaknesses:**

1. The research perspective of Proposition 3.3 seems interesting. The authors conduct research on temporal heterogeneity and gradient allocation problems for different time-steps.

2. As shown in Tab.2 and Tab.4, the proposed method seems to be more effective for neuromorphic datasets (e.g. DVS-CIFAR10), but the performance improvement on ImageNet-1k (large-scale dataset) and SNN Transformer architecture appears to be relatively limited (<1%). For CIFAR-100 dataset, the reported accuracy of this work lags behind TCL and ETC.

**Questions For Authors:**

See Strengths And Weaknesses Section.

**Relation To Broader Scientific Literature:**

The research field of this work is related to the BPTT training methods based on surrogate gradients in the SNN community.

**Theoretical Claims:**

I have read the theoretical claims in Sec. 3.2-3.3 and Appendix.

---

> ### Author Rebuttal · Authors · 2025-04-01
>
> **Rebuttal Appendix:https://anonymous.4open.science/r/TMC-0262**
>
> **1. I tend to think the proposed method can provide a new perspective for the SNN community to a certain extent. The research perspective of Proposition 3.3 seems interesting. The authors conduct research on temporal heterogeneity and gradient allocation problems for different time steps.**
>
> We sincerely appreciate your feedback and are greatly encouraged by it.
>
> **2. As shown in Tab.2 and Tab.4, the proposed method seems to be more effective for neuromorphic datasets (e.g. DVS-CIFAR10), but the performance improvement on ImageNet-1k (large-scale dataset) and SNN Transformer architecture appears to be relatively limited (<1%). For CIFAR100 dataset, the reported accuracy of this work lags behind TCL and ETC.**
>
> Here, we highlight the performance advantages of our TMC method on neuromorphic datasets, ImageNet and CIFAR100. Furthermore, we evaluate TMC on real-world applications and the results demonstrate that TMC **achieves SOTA performance**.
>
> **1) Neuromorphic Datasets**
>
> + **Dataset Feature Description:** These datasets capture the dynamic changes in pixel intensity and record the resulting spike events using dynamic vision sensors (DVS). Compared to static or frame-based datasets, they contain **rich spatio-temporal components** by interacting the spatial and temporal information and follow the event-driven processing fashion triggered by binary spikes.
> + **TMC Advantages:** Our method employs temporal heterogeneous learning using a temporal logit gradient rescaling strategy. This is beneficial for **capturing more task-relevant spatio-temporal dynamics features** compared to existing temporal homogeneous methods, such as SDT and TET. Therefore, TMC exhibits significant advantages on neuromorphic datasets.
>
> **2) ImageNet Dataset**
>
> + **Dataset Feature Description:** ImageNet is known for its high-resolution and complex scenes. Current efforts to enhance the performance of ImageNet mainly concentrate on refining the model architecture, often resulting in the **introduction of a large number of additional parameters** to achieve performance gains.
> + **TMC Advantages:** It is notable that our proposed method TMC is **plug-and-play**. When TMC is integrated into the Hierarchical Spiking Transformer (current SOTA model) in our paper, it enhances both classification and calibration performance on ImageNet, further demonstrating the effectiveness of our method.
>
> **3)CIFAR100 Dataset**
>
> + **Dataset Feature Description:** CIFAR-100 is characterized by its low-resolution images, which are often blurry and contain relatively less information.
> + **TMC Advantages:** With a small number of time steps, i.e., **T=2**, our method, which employs a temporal heterogeneous learning approach, is capable of capturing a richer set of features. Specifically, when T=2, TMC achieves a classification accuracy of 76.35%, outperforming other methods such as TEBN (75.86%), RMP-Loss (74.66%), and ETC (75.96%).
> + **Limitation of the Dataset:** However, when **the number of time steps is increased**, our method may capture noisy features due to the blurriness of the images, which is not beneficial to performance improvement. This suggests that our method has more advantages when handling dynamic and complex tasks.
>
> **4) Real-Word Application**
>
> Moreover, we evaluate our method on real-world applications, including text sequential classification and dynamic action sequential recognition tasks, where **TMC achieves state-of-the-art (SOTA) performance**.
> + **Text Sequential Classification Task.** We apply our method in the direct training phase of SpikingBert model proposed in [R1] on the widely used dataset Quora Question Pair (QQP). We compare the classification performance of TMC with SDT (as used in the original paper) and TET. TMC achieves a higher accuracy of **87.86\%** than SDT(86.82\%) and TET(87.03\%).
> + **Dynamic Action Sequential Recognition Task.** We train VGGSNN with T=10 on DVS-Gesture and compare the performance of our method with current works in Table R1 of Rebuttal Appendix. Results show that TMC achieves SOTA performance with an accuracy of 99.12%.
>
> **3.We futher evaluate the training process of TMC on VGGSNN on DVSCIFAR10 with T=10.**
>
> **1)Gradient Norm Evaluation:**
>
> In Figure R3 in the appendix, we evaluate the trend of gradient norms for model parameter updates during the training of TMC, SDT, and TET. TMC exhibits the highest norm, indicating **faster convergence**.
>
> **2)Training Stability Evaluation:**
>
> We visualize the training loss variation during the training of TMC, SDT and TET in Figure R4 in the appendix. The results demonstrate TMC's **stable training, driving loss to minimal values**, while SDT and TET suffer from overconfidence-induced oscillations.
>
>
> [R1] Spikingbert: Distilling bert to train spiking language models using implicit differentiation.

---

### Official Review · Reviewer_KZ4r · 2025-03-06

**Overall Recommendation:** 3

**Summary:**

This paper proposes a temporal confidence calibration method for SNNs, improving both the model's performance and heterogeneity, as demonstrated across several static classification tasks.

**Claims And Evidence:**

The proposed method is simple yet efficient, with a clear motivation at increasing the model's temporal heterogeneity. Comprehensive Experiments on static datasets clearly demonstrate its superiority over SDT and TET. The SOTA results on CIFAR10DVS and Imagenet are compelling.

**Essential References Not Discussed:**

No

**Experimental Designs Or Analyses:**

The experiments are conducted fairly and reasonably.

**Methods And Evaluation Criteria:**

1. The presented comparative and ablation experiments make sense for evaluating the method's effectiveness.

2. Evaluating only static datasets is fair yet a little weak regarding a method focusing on temporal heterogeneity. I hope the authors can provide more results on dynamic datasets to make the method more convincable, like action recognition datasets including DVS-Gesture, SL-Animals, etc.

**Other Comments Or Suggestions:**

No

**Other Strengths And Weaknesses:**

1. Some sentences in section 3.1 are false. [1] Calculate Loss on the output spikes. TET calculates the loss on the output values of an MLP followed by a spike layer.

2. The definition of the confidence score is quite weird, not bounded in [0, 1]. Leveraging it to punish overconfidence seems far deviated from the confidence calibration motivation.

3. Following the above, I didn't see much relationship between the proposed method and Definition 3.1, nor a relationship between Definition 3.1 and 3.2. The paper needs careful revising regarding the writing.

4. The result on N-Caltech101 is not SOTA. See [3].

[1] Spatio-Temporal Backpropagation for Training High-Performance Spiking Neural Networks

[2] Temporal efficient training of spiking neural network via gradient re-weighting

[3] Rethinking the membrane dynamics and optimization objectives of spiking neural networks

**Questions For Authors:**

See above

**Relation To Broader Scientific Literature:**

It could benefit the brain-inspired learning area a lot, especially the training of classification tasks that leverage cross-entropy loss on spiking neural networks.

**Theoretical Claims:**

In definition 3.1, is $\hat{P}_t$ the maximum value of softmax on the time-averaged logits? If so, the accuracies of the SNN at all time steps are equal. Then how does a temporally perfectly calibrated model trained with gt increase its temporal confidence monotonically to match Definition 3.1? (lines 199-202)

---

> ### Author Rebuttal · Authors · 2025-04-01
>
> **Rebuttal Appendix:https://anonymous.4open.science/r/TMC-0262**
>
> **1.Provide more results on dynamic datasets to make the method more convincable, like action recognition datasets including DVS-Gesture, SL-Animals, etc.**
>
> **1)DVS-Gesture**: We train VGGSNN with T=10 on DVS-Gesture and compare the performance of TMC with current works in Table R1 in the appendix. TMC achieves **SOTA performance with an accuracy of 99.12%**.
>
> **2)SL-Animals-DVS**: VGGSNN (instead of a larger model due to time constraints) is trained using TMC with T=16. Comparison with SDT and TET shows TMC achieving a **higher accuracy of 70.05%** against SDT (66.75%) and TET (68.34%), demonstrating the superiority of TMC.
>
> **2.Is $\hat{P}_t$ the maximum value of softmax on the time-averaged logits? Then how does a temporally perfectly calibrated model trained with gt increase its temporal confidence monotonically to match Definition 3.1?I didn't see much relationship between the proposed method and Definition 3.1, nor a relationship between Definition 3.1 and 3.2.**
>
> **Clarification of Definition 3.1 and Our Mechanism:**
>
> **1)Element Definition(lines 165-169 left)**
>
> + Temporal Predicted Confidence: $\hat{P}_t=max(Softmax(\overline{z_t}))$.
> + Accumulated Logit Outputs at t-th time step: $\overline{z_t}=\frac{1}{t}\sum_i^tz_i$.
> + $t \in {[1,2,...,T]}$.
> + **$\hat{P}_t$ is different at different time steps**, because $\overline{z_t}$ varies across time steps.
>
> **2)Definition 3.1:Temporally Perfectly Calibrated SNN(lines 169-176 left)**
>
> + Introduce model calibration into SNN's time dimension, where a temporally perfectly calibrated SNN satisfies **Test Accuracy is Equal to Confidence at each timestep**: ${\hat{\textit{P}_t}} = \mathbb{P}(\hat{\textit{y}}=\textit{y}|\hat{\textit{P}_t})$.
>
> **3)Definition 3.2:Temporal Gradient Scaling Factor(lines 177-176 left)**
>
> Model calibration is realized using a gradient rescaling factor to rescale the gradient of cross-entropy(CE) loss in the training of ANNs. To realize the temporal model calibration in SNN, we propose the temporal gradient scaling factor $g_t$ (lines 177-180 left) to rescale the gradient of CE loss at each time step, i.e., TET loss, in Equation 6.
>
> **4)Effect of $g_t$**
>
> + Note that Definition 3.1 constructs the concept of temporally perfectly model calibration in the field of SNN. Our work focuses on the rate-coding SNNs and we provide a further instantiation specific to rate-coding SNNs (lines 196-202 left). Specifically, current work suggests that the accuracy of rate-coding SNNs increases monotonically with time steps. Thus, for the temporally perfectly rate-coding SNNs, confidence should increase monotonically with time steps.
> + To match definition 3.1, $g_t$ should meet: at earlier time steps, it should shrink the effect of CE loss logit gradient ($\Delta Z_t^{TET}$) to reduce the confidence. Conversely, $g_t$ should enhance the effect of $\Delta Z_t^{TET}$ to increase the confidence.
>
> **5)Method Design (Section 3.3)**
>
> In Section 3.3, we propose TMC loss function with a new regularization term to realize the effect of $g_t$. With temporal gradient theoretical analysis (lines 244-274 left and 220-235 right), we derive the temporal gradient rescaling factor $g_t^{TMC}$ in our method, which can optimize the rate-coding SNN to increase its confidence monotonically with time steps.
>
> **3.Some sentences in section 3.1 are false.**
>
> + Thank you for your correction. We follow the loss function definition of SDT in [R13] and the reference [1] in line 139 left should be changed to [R13]. [R13]  defines the loss function of SDT by calculating the cross-entropy loss between the average pre-synaptic input of the output layer and the true label.
> + TET calculates the cross-entropy loss between the pre-synaptic inputs of the output layer with the true labels at each time step. To be more accurate, we should change "the membrane potential of the last layer" in our paper to "pre-synaptic input of the output layer".
>
> **4.The definition of the confidence score is quite weird, not bounded in [0, 1].**
>
> We would like to provide clarification regarding the "confidence score" in our paper.
>
> + Before Equation 9, our analysis focuses on absolute confidence values, that is, the maximum predicted probability, which is inherently bounded within the interval [0, 1].
> + For Equation 9, to enhance the loss function's sensitivity to overconfidence, we adopted the ratio of confidence/(1 - confidence) as $\theta_t$. This essentially serves as a more effective regularization of confidence.
>
> **5.Result on N-Caltech101 is not SOTA.**
>
> On N-Caltech101, our method achieves 86.03% accuracy at T=10, while [3] achieves 87.86% accuracy at T=16. **Reevaluating our method at T=16, we achieve 88.24% accuracy, surpassing [3]**.
>
> [R13] Deng, S.,Li,Y.,Zhang, S., andGu, S. Temporal efficient training of spiking neural network via gradient re-weighting. arXivpreprintarXiv:2202.11946,2022.

---

> > ### Comment · Reviewer_KZ4r · 2025-04-04
> >
> > Thanks for the author's rebuttal. It greatly helps me understand the paper more thoroughly. I am satisfied with the good results on both object recognition and action recognition results. I have one last question:
> >
> > Could you provide any theoretical or experimental analysis on whether the temporal confidence of a model trained with TMC could converges to its accuracy, in consistent with definition 3.1?

---

> > > ### Author Response · Authors · 2025-04-04
> > >
> > > Thank you for your review and positive feedback on our results in object recognition and action recognition. Regarding your new question, we provide both theoretical and experimental analysis as follows:
> > >
> > > **1. Theoretical Analysis**
> > >
> > > **1.1. Definition 3.1 requires a rate-coding SNN to satisfy the following two properties:**
> > > + **Property 1:** ${\hat{\textit{P}_t}} = \mathbb{P}(\hat{\textit{y}}=\textit{y}|\hat{\textit{P}_t})$.
> > > + **Property 2:**  $\hat{P_t} < \hat{P_{t+1}}$.
> > >
> > > Here, $\hat{P}_t=max(Softmax(\overline{z_t}))$ and $\overline{z_t}=\frac{1}{t}\sum_i^tz_i$.
> > >
> > > **1.2. Convert the realization of these two properties into the optimization objective of TMC:**
> > > + **Note:** Since the predicted outputs of a trained SNN typically have the highest probability for the target class across time steps, $\hat{P}_t$ can be expressed as  $\hat{P}_t=\overline{P}_t^k$ (lines 165-189 right). Here, $\overline{P}_t^k$ is the probability of the target class $k$ in the distribution of $Softmax(\overline{z_t})$.
> > >
> > > + **Objective 1:** During training, the realization of Property 1 can be converted to optimize $|\overline{P}_t^k - \mathbb{P}(\hat{\textit{y}}=\textit{y}|\overline{P}_t^k)| < \epsilon$. This can be achieved by introducing confidence regularization term, $\theta_t$ (lines 198-211 right), to penalize the under-confidence issue $\overline{P}_t^k < \mathbb{P}(\hat{\textit{y}}=\textit{y}|\overline{P}_t^k)$  and, especially, the over-confidence issue $\overline{P}_t^k > \mathbb{P}(\hat{\textit{y}}=\textit{y}|\overline{P}_t^k)$ with high sensitivity.
> > >
> > > + **Objective 2:** During training, the realization of Property 2, $\hat{P_t} < \hat{P_{t+1}}$, can be converted to $\overline{P_t}^k < \overline{P_{t+1}}^k$. This can be achieved by introducing a linearly decreasing exponent, $\lambda_t$ (lines 213-219 right and lines 220-223 left), into $\theta_t$ to optimize $z_t^k < z_{t+1}^k$, as described in Proposition 3.3 (lines 189-193 right).
> > >
> > > **1.3. Theoretical analysis of TMC's gradient rescaling factors:**
> > > + With the loss function of TMC, the rescaling factor for the target class $k$ is generated to optimize confidence (lines 243-254 left). Specifically,
> > > \begin{equation}g_t^k=\frac{\Delta Z_t^{TMC}}{\Delta Z_t^{TET}}=1-f(t) * h(t),~~~f(t)=\frac{\lambda_t\theta_t^{\lambda_t}}{t},~~~h(t)=\frac{1}{1-P_t^k}.\tag{R5}\end{equation}
> > > Here, $f(t)$ decreases with time steps.
> > > + **At the initial training phase**, $h(t)$ follows a random uniform distribution, and $g_t^k$ increases with time steps to optimize $z_t^k < z_{t+1}^k$, thereby meeting **Objective 2**.
> > > + **During training**, at time step $t$, if $z_t^k$ is high, the probability of the target class $k$ in the distribution of $Softmax(z_t)$, denoted as $P_t^k$, may lead to overconfidence. In this situation, $h(t)$ increases, causing $g_t^k$ to decrease potentially even to a negative value, to penalize the overconfidence issue. Conversely, underconfidence occurs when $z_t^k$ is low, and $g_t^k$ increases to address this issue, thereby meeting **Objective 1**.
> > > + **At the end of the training**, $g_t^k$ for different samples will converge to an interval. We have visualized the distribution of $g_t^k$ values for 500 samples of a trained SNN in **Figure 1 (detailed analysis can be found in lines 246-272 right)**. Notably, for most samples with reasonable confidence, their $g_t$ values are centered within an interval that shifts closer to 1 over time. This indicates the achievement of **Objective 2**. It can be seen that across time step, some samples' $g_t^k$ values are close to 1 or negative numbers. This is consistent with **Objective 1**, which aims to penalize particularly underconfident and overconfident samples. **Overall, TMC has the effect of realizing these two optimization objectives and further realizing Definition 3.1**.
> > >
> > > **2. Experimental Evaluation**
> > >
> > >  **2.1. Evaluation Metrics:** Whether the temporal confidence of a model trained with TMC could converge to its accuracy can be quantified by the calibration performance. Specifically, we evaluate the **calibration errors** (differences between the model's predicted confidence and its actual accuracy) of the trained VGGSNN (T=10) by TMC on DVSCIFAR10 across all time steps using the standard metrics ECE and AdaECE (detailed definition is in Appendix A.2 in our paper).
> > >
> > > **2.2. Experimental Results:** We compare the performance of TMC with SDT and TET in Table 1 in the appendix ( https://anonymous.4open.science/r/TMC-0262/Table1_Calibration_Performance_Results.pdf). TMC exhibits the lowest calibration errors across time steps, indicating **the model's predicted confidence is optimized to converge to accuracy.**
> > >
> > > **2.3. Experiment in Paper:** We compare overall calibration performances on different datasets in **Table 2 (lines 330-340 left)**. TMC achieves the lowest calibration errors.
> > >
> > > **If there are any questions, please let us know. We would also greatly appreciate any consideration for a minor adjustment in the rating.**

---

### Official Review · Reviewer_GDhn · 2025-03-13

**Overall Recommendation:** 2

**Summary:**

this paper introduces a new training method for spiking neural networks called temporal model calibration . the goal is to improve the performance of snns by increasing their temporal heterogeneity, which is how much their outputs vary over time. the authors argue that existing training methods, like direct training using bptt, do not fully utilize temporal heterogeneity because the loss gradients remain too similar across time steps. their main idea is to rescale the loss gradients at each time step to encourage more diversity in the network’s responses over time. they do this by modifying the cross-entropy loss function with a new gradient scaling factor based on confidence calibration techniques used in deep learning. the method is tested on several datasets, including imagenet, dvscifar10, and n-caltech101, and achieves state-of-the-art accuracy in some cases. overall, the paper presents a novel way to enhance the learning dynamics of snns by focusing on how gradients evolve over time.

## update after rebuttal

Thanks to the authors for a very detailed and thoughtful response. I appreciate the new experiments, the visualizations of temporal heterogeneity, and the extra analysis around computational cost and training stability — it’s clear a lot of effort went into the rebuttal. These additions definitely helped clarify several points I had raised.

That said, my overall opinion on the paper hasn’t changed much. While the new results are nice to see, they don’t fully address my bigger concerns about the lack of strong theoretical backing and the limited discussion on scaling to larger models and broader applications. The paper is solid and the method works well empirically, but I still feel it falls a bit short on the novelty and depth needed for acceptance. So I’m keeping my original score of Weak Reject.

**Claims And Evidence:**

the main claim of the paper is that current direct training methods for snns do not make full use of temporal heterogeneity because their loss gradients remain relatively uniform across time steps. the authors support this claim by analyzing the gradients in standard training methods and showing that they lack diversity. they further claim that their proposed tmc method improves both temporal heterogeneity and accuracy by rescaling these gradients. the experimental results provide reasonable support for this, as the tmc-trained models outperform existing methods on several benchmark datasets. however, the claim that the method enhances temporal heterogeneity is mostly based on indirect evidence (such as improved accuracy) rather than direct visualization or mathematical proof of increased heterogeneity. a more in-depth analysis of how tmc affects neuron activations over time would strengthen this claim.

**Essential References Not Discussed:**

the paper includes most of the key references in snn training and model calibration but does not discuss some recent works on improving training stability through adaptive gradient methods. for example, methods that dynamically adjust learning rates based on gradient variance might have some similarities to tmc. comparing tmc to these approaches could provide additional insights into its uniqueness and potential limitations.

**Experimental Designs Or Analyses:**

the experimental design is solid in terms of dataset selection and performance metrics. the authors compare their method against several baselines, including standard direct training (sdt) and temporal efficient training (tet), which are well-known methods in the field. they show consistent improvements in accuracy and calibration errors across multiple datasets. one strength of the experiments is that they include both static and neuromorphic datasets, showing that tmc is broadly applicable. however, there are some missing analyses. for example, the paper does not include an ablation study to determine which components of tmc contribute most to the performance gains. it would be useful to see experiments testing different variations of the gradient rescaling factor to understand its specific effects. also, the paper does not analyze how sensitive tmc is to hyperparameter choices, which is important for practical use.

**Methods And Evaluation Criteria:**

the proposed method is well-designed for the problem it addresses, as it directly targets the temporal structure of snns. the authors evaluate their approach using standard benchmark datasets, including imagenet and neuromorphic datasets like dvscifar10 and n-caltech101. these datasets are appropriate for testing the effectiveness of an snn training method. the evaluation primarily focuses on accuracy, expected calibration error (ece), and adaptive ece, which are relevant metrics for both classification performance and model calibration. however, the paper does not provide much discussion on computational efficiency—how much additional training time or memory tmc requires compared to existing methods. since gradient rescaling could introduce extra computation, it would be useful to see an analysis of the trade-offs between performance and efficiency.

**Other Comments Or Suggestions:**

it would be helpful to include a computational cost analysis comparing the training time of tmc with other methods. a more detailed discussion of hyperparameter sensitivity would also improve the paper. the writing is mostly clear, but some sections could be better structured to make the key ideas easier to follow.

**Other Strengths And Weaknesses:**

one of the main strengths of this paper is that it introduces a relatively simple yet effective modification to the loss function that improves performance across multiple datasets. the empirical results are strong, showing state-of-the-art accuracy in some cases. another strength is that the paper addresses an important issue in snn training—the lack of sufficient temporal heterogeneity—by introducing a novel way to enhance it. however, there are some weaknesses. the paper does not provide a theoretical justification for why tmc should always improve training, and there is little discussion of potential downsides, such as increased training time or instability. also, while the results are impressive, the authors do not explore the scalability of tmc to much larger models or real-world applications.

**Questions For Authors:**

1. how much additional computational overhead does tmc introduce compared to standard direct training methods? does it significantly increase training time?
2. have you tested whether tmc works well with much deeper networks or more complex architectures?
3. do you have any direct visualizations of how tmc changes the temporal activity of snn neurons over time? this could provide stronger evidence that it increases temporal heterogeneity.
4. is there any risk that rescaling the gradients could introduce instability in training, such as oscillations or divergence? if so, how is this mitigated?

**Relation To Broader Scientific Literature:**

this paper builds on prior work in spiking neural networks, particularly methods that use backpropagation through time for training. it connects to research on temporal heterogeneity in snns, which has been studied in both computational neuroscience and machine learning. the work is also related to research on model calibration in deep learning, as tmc is inspired by techniques like label smoothing and confidence regularization. while the paper does a good job of citing relevant works, it does not compare tmc to alternative approaches that also modify loss gradients for better learning dynamics, such as entropy-based loss functions. discussing these related methods could provide a clearer context for how tmc fits within existing strategies.

**Theoretical Claims:**

the paper provides a detailed mathematical formulation of the proposed method, including how the gradient rescaling factor is computed. however, there is no formal proof that tmc leads to a more stable or optimal training process. the method is motivated by intuition and empirical results rather than rigorous theoretical guarantees. for example, while the authors argue that their gradient rescaling improves learning dynamics, they do not analyze whether it guarantees faster convergence or prevents issues like vanishing or exploding gradients. a theoretical analysis of the convergence properties of tmc would make the claims more robust.

---

> ### Author Rebuttal · Authors · 2025-04-01
>
> **Rebuttal Appendix:https://anonymous.4open.science/r/TMC-0262**
>
> **1.Visualization or mathematical proof of increased heterogeneity.In-depth analysis of how tmc affects neuron activations.**
>
> **1)Temporal Heterogeneity Visualization**
>
> We compare VGGSNNs trained by TMC and TET on DVSCIFAR10, visualizing the cosine similarity of layer features across time steps in Figure R1. TMC exhibits higher temporal heterogeneity at all layer levels.
>
> **2)TMC Effect**
>
> The neuron model:
>  \begin{equation} {u_{t+1}^i}=\lambda(u_t^i-V_{th}s_t^i)+\sum_j\mathbf{W_{ij}}s_{t+1}^j.\tag{R4}\end{equation}
>
> With TMC $\mathbf{W}$'s separability for varied inputs is enhanced, boosting neuron temporal heterogeneity in response to diverse temporal inputs.
>
> **2.Discussion on computational efficiency.**
>
> We analyze TMC's time and space complex of the logit gradient calculation relative to SDT and TET. (T:Time Step, B:Batch Size, C:Class Number)
>
> **1)SDT**:Time complexity $O(T\*B\*C)$, space complexity $O(B\*C)$.
>
> **2)TET**:Time complexity $O(T\*B\*C)$, space complexity $O(T\*B\*C)$.
>
> **3)TMC**:Based on TET, it introduces additional calculation of the regularization term $\theta_t^{\lambda_t}$. At each timestep, generating $\theta_t$ needs $O(B\*C)$ time complexity and $O(1)$ space complexity. Calculating $\theta_t^{\lambda_t}$ needs $O(1)$ for both time and space complexity. Overall, TMC's logit gradient calculation time complexity is $O(T\*B\*C)+O(T\*B\*C)+O(T\*B\*C)+O(T)$, and space complexity is $O(T\*B\*C)+O(T\*B\*C)+O(T)+O(T)$.
>
> TMC's total gradient computation, including hidden layer and logit gradients, is **close to SDT and TET in computation complexity and training time**.
>
> **3.No formal proof that tmc leads to a more stable or optimal training.Theoretically analyze the convergence properties of tmc.**
>
> **1)Firing Rate Evaluation:**
>
> Firing rates impact SNN gradients. As shown in Figure R2, compared with SDT and TET, TMC-trained VGGSNN on DVS-CIFAR10 exhibits the lowest firing rates in shallow layers, rising with depth, and peaking in deeper layers. This indicates TMC's sensitive response to global features in deeper layers. Higher firing rates also **mitigate the gradient vanishing**.
>
> **2)Gradient Norm Evaluation:**
>
> Figure R3 further compares model parameters update gradient norms of TMC, SDT and TET. TMC shows the highest norm, suggesting **faster convergence**.
>
> **3)Training Stability Evaluation:**
>
> We theoretically analyze TMC's temporal rescaling factor $g_t$(lines 224-274 left). $g_t$ effectively rescale logit gradients, enhancing optimization over SDT and TET. Figure R4 demonstrates TMC's **stable training**, driving loss to minimal values, unlike SDT and TET, which suffer from overconfidence-induced oscillations.
>
> **4.The paper does not include an ablation study to determine which components of tmc contribute most to the performance gains.Analyze hyperparameter sensitivity of tmc.**
>
> 1)In lines 358-384 left and 330-336 right, **we conducted ablation studies** on the regularization term's components—base $\theta_t$ and exponent $\lambda_t$. Both enhance performance, but their combined effect is optimal.
>
> 2)TMC **does not introduce additional hyperparameters**, ensuring flexibility.
>
> **5.It does not compare tmc to alternative approaches.It does not discuss some recent works on improving training stability through adaptive gradient methods.**
>
> **1)Compare to Alternative Approaches:** In Related Work section and lines 149-155 left, we discussed entropy-based loss function methods and highlighted their homogeneous training effect. Notably, TMC has a different training effect by introducing temporal heterogeneous training. As the code for these methods is not available, we can only compare TMC's classification accuracy with them in Section 4.4.
>
> **2)Compare to Adaptive Gradient Methods:** Recent studies[R10-R12] have introduced adaptive gradient methods, primarily focusing on adjusting hidden layer gradients and using SDT loss to calculate the logit gradient. In contrast, TMC modifies logit gradient calculations and offers plug-and-play compatibility with various hidden layer gradient mechanisms.
>
> **6.Explore tmc to much larger models,real-world applications,deeper networks or more complex architectures.**
>
> **1)Larger Model Evaluation:** Hierarchical Spiking Transformer used in our paper with 64.96M parameters is the large model.
>
> **2)Deeper Model Evaluation:** We verify TMC on ResNet101 on ImageNet dataset with T=4 and compare it with SDT and TET. TMC achieves a higer accuracy of **70.52%** than SDT(68.74%) and TET(67.98%).
>
> **3)Real-World Applications:** We evaluate TMC on text sequential classification and dynamic action sequential recognition tasks and TMC achieves **SOTA performance**. More detailed results can be seen in Response 3 to Review ksZk.
>
> [R10]Learnable Surrogate Gradient for Direct Training Spiking Neural Networks
> [R11]Sparse Spiking Gradient Descent
> [R12]Gradient Descent for Spiking Neural Network

---

### Official Review · Reviewer_ksZk · 2025-03-14

**Overall Recommendation:** 3

**Summary:**

This work finds that the logit gradients have insufficient diversity in the temporal dimension during SNN training. The authors then rescale the gradient in each time step to improve diversity, resulting in SOTA performance for image classification tasks.

**Claims And Evidence:**

yes

**Essential References Not Discussed:**

None

**Experimental Designs Or Analyses:**

checked

**Methods And Evaluation Criteria:**

yes

**Other Comments Or Suggestions:**

Suggestions:

1. Given the focus on temporal heterogeneity, I believe that sequential tasks would serve as more appropriate benchmarks to be considered.

2. The importance of temporal heterogeneity should be clarified.

**Other Strengths And Weaknesses:**

Strengths:
1. The method is in a plug-and-play fashion and can be merged with most SOTA models and algorithms.
2. It achieves SOTA performance on common image classification tasks.

Weaknesses:
1. Lack of novelty. The method can be regarded as a minor modification of TET: it is like a cumsum version of TET. Even the regularization term also exhibits a comparable effect to the $L_{MSE}$ regularization in TET.

2. The logic is not convincing. While marginal modifications can indeed constitute valuable contributions, this paper fails to show the significance of such marginal changes clearly. It is not clear why diversity across time should be improved. And the regularization to improve diversity is not straightforward.

**Questions For Authors:**

Interestingly, the proposed method works better on neuromorphic datasets than static images. Do the authors consider this a common phenomenon or not? Could the authors explain the possible reason? Are there fundamental insights within this observation?

**Relation To Broader Scientific Literature:**

This work improves TET by increasing logit diversity in the temporal dimension.

TET: Temporal efficient training of spiking neural network via gradient reweighting

**Theoretical Claims:**

The technical detail is limited:

1. It is not clear whether the so-called "logit gradient" should be diverse across time steps. The surrogate gradient + firing rate-based loss framework (both SDT and TET) implicitly follows a rate coding scheme. Consequently, it might be that uniformity in behavior over time steps serves as a more reliable indicator of confidence. The relationship between temporal heterogeneity and the performance of SNNs remains unclear and needs a formal theoretical explanation.

---

> ### Author Rebuttal · Authors · 2025-04-01
>
> **Rebuttal Appendix:https://anonymous.4open.science/r/TMC-0262**
>
> **1.The relationship between temporal heterogeneity and the performance of SNNs remains unclear. It is not clear whether the so-called logit gradient should be diverse.**
>
> **1)Relationship between Temporal Heterogeneity and Performance**
>
> In SNNs, the neuronal dynamics of the membrane potential $u_t$ can be formulated as:
>  \begin{equation} {\tau\frac{du_t}{dt}}=-(u_t-u_{reset})+I_t.\tag{R1}\end{equation}
> When $u_t$ reaches the threshold $V_{th}$, $u_t$ is set to $u_{rest}$ and the neuron fires a spike:
> \begin{equation} s_t=\sum_{t_f}\delta(t-t_f).\tag{R2}\end{equation}
> The temporal heterogeneity of SNNs implies $|\frac{du_t}{dt}|>0$, yielding benefits as follows:
> + **Reduction of Dead Neurons**: Ensuring neurons spike when necessary.
> + **Sensitive Response** to Temporal Dependency and Input Information: Capturing spatio-temporal dynamics effectively.
> + **Diverse Spike Firing Frequency**: Adjustable $\Delta t$ in $s_t$ allows for diverse behaviors, such as burst spikes.
>
> Overall, accumulating temporal heterogeneous neuron responses to varied temporal inputs over time steps **enhances predictive confidence and performance**.
>
> **2)Temporal Heterogeneity Improvement**
>
> Direct training of SNNs involves the discrete model expression:
>  \begin{equation} {u_{t+1}^i}=\lambda(u_t^i-V_{th}s_t^i)+\sum_j\mathbf{W_{ij}}s_{t+1}^j.\tag{R3}\end{equation}
> Enhancing temporal heterogeneity can be achieved by improving either the temporal dependency dynamics $\lambda(u_t^i-V_{th}s_t^i)$ or the input mapping dynamics $\sum_j\mathbf{W_{ij}}s_{t+1}^j$ via $\mathbf{W}$ training. **We concentrate on the latter, which is less explored.**
>
> **3)Logit Gradient Diversity Rationality**
>
> Enhancing linear separability, that is, increasing the rank of $\mathbf{W}$ is crucial for dynamic input mapping. Increasing temporal gradient diversity is beneficial to explore the solution space and increase $\mathbf{W}$ rank. However, existing methods lack exploration of logit gradient diversity.
>
>
> **2.The method is a minor modification of TET.This paper fails to show the significance of such marginal changes clearly.The regularization to improve diversity is not straightforward.**
>
> From the technological perspective, TMC is a regularization term modification based on TET. However, from the temporal gradient decent perspective, **TMC is a critical improvement of TET**. The significance of TMC is highlighted as follows:
>
> **1)Rethinking the Temporal Logit Gradient Calculation**
>
> + TET focuses on increasing the momentum of temporal gradient descent but ignores appropriately assigning magnitude and direction to the momentum. This leads to an overconfidence issue.
>
> + Introducing model calibration into SNNs, we rethink the temporal gradient calculation properties that SNNs should meet and propose a temporal gradient rescaling mechanism. This mechanism assigns appropriate magnitude and direction to the gradient descent momentum across time steps.
>
> **2)Effective and Adaptive Regularization Term**
>
> + TET introduces the MSE regularization to prevent the occurrence of "particular outliers" but lacks deeper analysis on the issues. The regularization term appears to be an intuitive design. Moreover, the regularization term at each time step keeps same and fixed, relying on hyperparameters.
> + During training, TMC with a new regularization term rescales the logit gradient effects of TET, adaptively responding to input data, training phases, and time steps without hyperparameter dependence.
> + There exist straightforward model calibration techniques like label smoothing, but these methods have been found to be suboptimal relying on prior information and hyperparameters. Moreover, the regularization-based calibration techniques achieve SOTA performance.
>
> **3.Sequential tasks are more appropriate benchmarks to be considered.**
>
> + **Text Sequential Classification Task.** TMC is applied during the direct training phase of the SpikingBert [R1] model on Quora Question Pair dataset. Compared to SDT(86.82%) and TET(87.03%), TMC achieves higher accuracy at 87.86%.
> + **Dynamic Action Recognition Task.** We train VGGSNN with T=10 on DVS-Gesture and compare the performance of TMC with current works in Table R1 in the appendix. TMC achieves SOTA performance with an accuracy of 99.12%.
>
> **4.TMC works better on neuromorphic datasets than static images. Provide explanation and insights.**
>
> **TMC indeed exhibits more significant advantages on neuromorphic datasets.**
> + Compared to static datasets, neuromorphic datasets have rich spatiotemporal components by interacting the spatial and temporal information.
> + TMC employs temporal heterogeneous training and has the advantage of capturing more task-relevant spatiotemporal dynamics features, outperforming existing methods, which employ temporal homogeneous training.
>
> [R1]Spikingbert: Distilling bert to train spiking language models using implicit differentiation.

---

> > ### Comment · Reviewer_ksZk · 2025-04-07
> >
> > 1. Still not convinced about the rationality of Temporal Heterogeneity.
> >
> > 2. DVS-Gesture is too simple and can be trained well with only static information.
> >
> > However, due to the great efforts the authors put in the rebuttal, the additional experiments, and the good overall performance of the proposed trick, I will raise my score.

---

> > > ### Author Response · Authors · 2025-04-07
> > >
> > > We would like to express our sincere gratitude for your consideration and the score increase. Here, we provide a detailed analysis of the rationality of temporal heterogeneity and more convincing experimental results.
> > >
> > > **1.Rationality of Temporal Heterogeneity**
> > >
> > > The behavior of SNN is determined by the input data and the basic neuronal processing units.
> > >
> > > **1) Definitions:**
> > >
> > > + **Input Data: Temporally Dynamical Sequence Data.** It includes two types: (1) Neuromorphic data. It naturally possesses temporal heterogeneity. (2) Static data. It exhibits temporal heterogeneity after being processed by a spiking encoding layer.
> > > + **Basic Neuronal Processing Units: Vanilla LIF Neurons.** The expression of the LIF neuron over continuous time is as follows:
> > > \begin{equation} {\tau\frac{du_t}{dt}}=-(u_t-u_{reset})+I_t.\tag{R1}\end{equation}
> > > When $u_t$ reaches the threshold $V_{th}$, $u_t$ is set to $u_{rest}$ and the neuron fires a spike:
> > > \begin{equation} s_t=\sum_{t_f}\delta(t-t_f).\tag{R2}\end{equation}
> > >
> > > **2) Neuronal Temporal Heterogeneity Analysis**
> > >
> > > Firstly, the change in neuronal membrane potential is quantified by $\frac{du_t}{dt}$, which is determined by two components:
> > > + **Decay of Membrane Potential $-(u_t-u_{reset})$.** This term indicates that the membrane potential naturally decays towards the resting potential $u_{reset}$. If $u_t > u_{reset}$, this term is negative, indicating a decrease in membrane potential. If $u_t < u_{reset}$, this term is positive, indicating an increase in membrane potential.
> > > + **Input Current $I_t$.** It represents the effect of external current on the membrane potential. If $I_t$ is positive, it drives the membrane potential up. If $I_t$ is negative, it pushes the membrane potential down.
> > >
> > > Secondly, we investigate the temporal dynamic changes of $\frac{du_t}{dt}$:
> > > + **When $\frac{du_t}{dt} > 0$**, it means the membrane potential $u_t$ is increasing. This could be due to an input current $I_t$ that is sufficiently large to overcome the natural decay of the membrane potential, or because the effect of the natural decay is relatively small at the current moment.
> > > + **When $\frac{du_t}{dt} < 0$**, the membrane potential $u_t$ is decreasing. This could be because the input current $I_t$ is small or negative, not enough to counteract the decay of the membrane potential, or the effect of the natural decay is relatively large at the current moment.
> > > + **Note that $\frac{du_t}{dt} = 0$**  is highly unlikely to occur unless specific conditions are met.
> > > + **Thus, $|\frac{du_t}{dt}| > 0$** indicates that the membrane potential is continuously changing, either increasing or decreasing. This change is a direct reflection of the neuron's response to temporal dependency and input signals and is **a key evidence of temporal heterogeneity.**
> > >
> > > **3) Neuronal Temporal Heterogeneity Improvement**
> > >
> > > To establish the computational link along the spatial-temporal dimension, the discrete vanilla LIF model can be formulated as:
> > >  \begin{equation} {u_{t+1}^i}=\lambda(u_t^i-V_{th}s_t^i)+\sum_j\mathbf{W_{ij}}s_{t+1}^j.\tag{R3}\end{equation}
> > > Neuronal dynamic can be reflected by $\lambda(u_t^i-V_{th}s_t^i)$ and  $\sum_j\mathbf{W_{ij}}s_{t+1}^j$.
> > >
> > >
> > > Existing research shows that **vanilla LIF neurons' responses to complex temporal sequence tasks are insufficient**. To enhance neuronal heterogeneity and boost SNN performance, two perspectives can be considered:
> > >
> > > + **Temporal Dependency Dynamic Enhancement.** Many works focus on this area, such as the proposed parametric spiking neurons.
> > > + **Input Response Dynamic Enhancement.** For complex tasks, effectively capturing the highly dynamical temporal features of input data is crucial for enhancing model performance.
> > >
> > > **4) Our Contribution**
> > > + The mapping function $W$ is key to capturing temporal features, and its dynamic response to varying inputs is critical for performance. Updating $W$ via logit gradient backpropagation through hidden layers should enhance its separability and sensitivity.
> > > + However, existing methods focus on improving hidden layer gradient backpropagation, while the logit gradient remains underexplored. Its limited diversity can hinder SNN performance. We aim to address this by **enhancing logit gradient diversity across time steps**, boosting $W$'s dynamic response, and capturing more dynamic information to improve performance.
> > >
> > > **2. More Convincing Experimental Results**
> > >
> > > **1) Deeper Model Evaluation:** We verify TMC on ResNet101 on ImageNet dataset with T=4 and compare it with SDT and TET. TMC achieves **a higer accuracy of 70.52%** than SDT(68.74%) and TET(67.98%).
> > >
> > > **2) Evaluation On SL-Animals-DVS:** VGGSNN (instead of a larger model due to time constraints) is trained with TMC with T=16. Comparison with SDT and TET shows TMC achieving **a higher accuracy of 70.05%** against SDT (66.75%) and TET (68.34%), demonstrating the superiority of TMC.

---

### Decision · Program_Chairs · 2025-05-01

**Decision:**

Accept (poster)

**Comment:**

This paper introduces *Temporal Model Calibration (TMC)*, a gradient rescaling strategy aimed at improving the temporal heterogeneity and overall performance of Spiking Neural Networks (SNNs). It proposes to rescale the cross-entropy loss gradients at each time step, thereby enhancing the diversity of logit gradients. The method is validated on a broad range of static and neuromorphic datasets, achieving competitive or state-of-the-art results.

**Strengths noted by reviewers include:**

- Simple yet effective design with no new hyperparameters and compatibility with a wide range of SNN architectures.
- Consistent performance improvement over SDT and TET on both static and neuromorphic datasets, including larger models like ResNet101 and SpikingBERT.
- Broad applicability validated through experiments on vision and sequential tasks, including DVS-Gesture and SL-Animals-DVS.
- Extensive rebuttal with detailed theoretical clarifications, new visualizations, calibration error evaluations (ECE/AdaECE), and computational complexity analysis.

**Concerns raised by reviewers and how they were addressed:**

- **Novelty and conceptual distinction from TET**: Reviewers noted that TMC appears to be an incremental improvement over TET. The authors argued that while TMC builds on TET, it introduces a principled calibration-based gradient rescaling mechanism that better handles confidence evolution over time.

- **Theoretical justification**: While no formal convergence proofs were provided, the authors offered analysis of the temporal rescaling factor, firing rate dynamics, and gradient norms, along with empirical evidence supporting faster and more stable convergence.

- **Calibration definition and clarity**: Reviewer KZ4r raised issues with Definitions 3.1 and 3.2. The rebuttal clarified the formulation and provided both theoretical objectives and empirical confirmation that temporal confidence converges to accuracy.

- **Limited gains on certain datasets**: Reviewer g8s8 noted smaller gains on ImageNet and CIFAR100. The authors responded with deeper analysis, showing TMC's strengths on dynamic or complex temporal tasks, and explained limitations due to low resolution or static nature of some benchmarks.

- **Visualization and interpretability**: Reviewer GDhn requested more direct evidence of increased heterogeneity. The authors provided cosine similarity plots across time steps, showing increased diversity in intermediate representations.

Overall, this paper offers a thoughtful and empirically solid contribution to SNN training, particularly in calibrating temporal dynamics via gradient rescaling. While the theoretical novelty remains somewhat incremental, the work is well-executed, addresses a relevant problem, and shows strong generalization across tasks and architectures. The rebuttal was detailed and resolved several key reviewer concerns.